# Hedonic Neurons: A Mechanistic Mapping of Latent Coalitions in Transformer MLPs

**Tanya Chowdhury, Atharva Nijasure, Yair Zick & James Allan**
Manning College of Information and Computer Sciences
University of Massachusetts Amherst
`{tchowdhury,anijasure,yzick,allan}@cs.umass.edu`

## Abstract

Fine-tuned Large Language Models (LLMs) encode rich task-specific features, but the form of these representations—especially within MLP layers—remains unclear. Empirical inspection of LoRA updates shows that new features concentrate in mid-layer MLPs, yet the scale of these layers obscures meaningful structure. Prior probing suggests that statistical priors may strengthen, split, or vanish across depth, motivating the need to study how neurons *work together* rather than in isolation.

We introduce a mechanistic interpretability framework based on *coalitional game theory*, where neurons mimic agents in a hedonic game whose preferences capture their synergistic contributions to layer-local computations. Using top-responsive utilities and the PAC-Top-Cover algorithm, we extract *stable coalitions of neurons*—groups whose joint ablation has non-additive effects—and track their transitions across layers as persistence, splitting, merging, or disappearance.

Applied to LLaMA, Mistral, and Pythia rerankers fine-tuned on scalar output tasks, our method finds coalitions with consistently higher synergy than clustering baselines. By revealing how neurons cooperate to encode features, hedonic coalitions uncover higher-order structure beyond disentanglement and yield computational units that are functionally important, interpretable, and predictive across domains.

## 1 Introduction

Consider a large language model fine-tuned to compute semantic similarity between text pairs. When presented with two sequences, the model outputs a scalar score, e.g. 0.76. But how is this number internally computed? Within the millions of parameters of a transformer, such decisions are not the work of isolated neurons, but of groups that cooperate to represent abstract features like "semantic overlap," "term frequency patterns," or "syntactic alignment." These computations may parallel familiar retrieval metrics – e.g., some neuron coalitions might compute TF-IDF-like statistics (Sparck Jones, 1972), others might capture cosine similarities between representations, while still others might encode position-dependent matching signals. Traditional interpretability methods have limitations here: probing (Gurnee et al., 2023; Hewitt & Manning, 2019) captures correlations with labels but ignores cooperation, sparse autoencoders (SAEs) (Huben et al., 2024) disentangle activations into monosemantic directions but overlook nonlinear dependencies, and clustering (Cao et al., 2025; Song et al., 2024) groups neurons by statistical proximity rather than functional interaction. What is missing is a principled way to identify *synergistic neuron groups*—subsets whose combined contribution exceeds the sum of their parts. We aim to identify the computational units that organize into stable coalitions, that potentially encode these mathematical concepts within scalar-output LLMs.

Recent work has shown that LoRA fine-tuning can teach LLMs new tasks by updating only mid-level MLP layers, nearly matching full fine-tuning (Hu et al., 2022; Zhou et al., 2024; Nijasure et al., 2025). Yet inspection of these LoRA weight updates reveals little obvious structure: millions of parameters diffuse across neurons, obscuring which units encode task-specific features. We hypothesize that the key to isolating LoRA emergent behaviour lies in identifying *coalitions* of neurons that consistently co-adapt under fine-tuning. Inspired by game theory, we model neurons as agents in a *hedonic game* (Dreze & Greenberg, 1980), where preferences reflect synergy with others. Though neurons are not literally rational, stochastic gradient descent imposes a form of selection pressure: directions that

reduce loss persist, and many neurons are only useful in combination (e.g., a feature computation). Thus, stable coalitions naturally emerge as groups of neurons that survive training together. This evolutionary analogy motivates the hedonic game framing: utilities capture how much a neuron's survival depends on its synergy with others, and stable coalitions correspond to groups of neurons that consistently co-adapt under training. By modeling these groups as coalitions, we open a path toward reverse-engineering their function and symbolically characterizing their emergent behavior.

*Why does this matter?* Beyond offering a new perspective on interpretability, coalition analysis provides actionable insight into how task-specific features are represented and evolve. By showing which neuron groups are functionally indispensable, our framework suggests future directions for practical interventions such as model comparison, transfer learning, or modular editing at the coalition level rather than at the level of individual weights. Moreover, tracking persistence, splits, and vanishings highlights how statistical priors are refined or discarded across depth, shedding light on the internal dynamics of fine-tuned models—information that clustering or SAE-style methods cannot reveal. Thus, stable coalitions are not only theoretically appealing but also open a path to understanding and eventually controlling the computational units that fine-tuning creates.

**Our Contributions**. We introduce a game-theoretic framework for discovering and analyzing neuron coalitions in transformer MLPs. (1) We model neurons as players in a hedonic cooperative game with top responsive utilities based on synergy, and solve for $\varepsilon$-PAC-stable outcomes using the PAC-Top-Cover algorithm. (2) We evaluate coalitions both intrinsically and extrinsically: compared to clustering baselines, they achieve +0.29 *Pairwise* and +0.49 *Ratio* synergy, exhibit 3–5$\times$ larger out-of-distribution performance drops under ablation, align more strongly with IR heuristics (BM25, IDF, query term coverage), and yield macro-features that improve predictive $R^2$ from $\sim$0.20 to 0.43–0.47. (3) Treating coalitions as "meta-neurons," we trace their evolution across consecutive layers, finding that most groups vanish or split while only a small fraction persist—supporting the view that deeper MLPs act primarily as feature filters rather than creators. Applied to LLaMA, Mistral, and Pythia LoRA rerankers, we show that hedonic coalitions consistently uncover reproducible and functionally indispensable computational units. To our knowledge, this is the first work to use game theory to identify, validate, and track synergistic neuron groups in fine-tuned LLMs. Details of all code, models, and datasets are provided here[1].

## 2 BACKGROUND

We begin by outlining the fundamentals of hedonic games and their application in modeling cooperative behavior. We then describe the transformer architecture with an emphasis on MLP sublayers.

### 2.1 HEDONIC GAMES AND PAC-STABLE COALITION FORMATION

A *hedonic coalition formation game* (Dreze & Greenberg, 1980) consists of a set of players $N$ who exhibit preferences over groups they might join. Formally, each player $i \in N$ ranks all coalitions $S \subseteq N$ that contain $i$; in our setting, we assume that players have *cardinal* utilities over coalitions. Given a player $i \in N$ and a coalition $S$ containing $i$, player $i$'s utility from joining $S$ is $u_i(S) \in \mathbb{R}$, which we later instantiate as a function of $i$'s strongest partners within $S$.

Our goal is to identify a *coalition structure* or *partition* of the player set which satisfies certain desiderata (Aziz & Savani, 2016). Given a coalition structure $\pi$, we let $\pi(i)$ designate the coalition containing player $i$ under $\pi$. *Core stability* (Bogomolnaia et al., 2002) is a key cooperative solution concept. We say that a coalition $S \subseteq N$ *blocks* a coalition structure $\pi$ if every player $i \in S$ strictly prefers $S$ to their assigned coalition $\pi(i)$, i.e., $u_i(S) > u_i(\pi(i))$ for all $i \in S$. A coalition structure $\pi$ is *core stable* (or simply *stable*) if no blocking coalitions exist.

Enumerating agents' preferences over all coalitions is infeasible; with $n$ agents, each agent needs to express their preferences over $2^{n-1}$ potential groups. Sliwinski & Zick (2017) propose using *Probably Approximately Correct (PAC)* guarantees (Kearns & Vazirani, 1995; Shashua, 2009). The key insight of this framework is to sample players' preferences rather than utilize complete preferences over all coalitions. Given a distribution $D$ over coalitions, a coalition structure $\hat{\pi}$ is called $\varepsilon$-*PAC stable* if

$$\Pr_{S \sim D}\big[S \text{ core blocks } \hat{\pi}\big] \leq \varepsilon.$$

---

[1] https://github.com/TaKneeAa/hedonicNeurons

Here, $D$ is the distribution over sampled coalitions used to approximate neuron preferences. Intuitively, while it is *possible* that $\hat{\pi}$ is not core stable, the probability of observing a blocking coalition for $\hat{\pi}$ under the distribution $D$ is small.

A PAC stabilization algorithm takes $m = \text{poly}(n, \frac{1}{\varepsilon}, \log \frac{1}{\delta})$ samples from $D$ and outputs an $\varepsilon$-PAC stable partition with probability at least $1 - \delta$. Intuitively, $\delta$ captures the probability that the $m$ samples we took are not representative of the 'true' data distribution $D$.

**Top-Responsive Hedonic Games in Neural Networks.** We estimate pairwise affinities $\phi_{ij}$ between "players" (neurons) from weights and co-activations. These affinities allow us to construct a hedonic game in which each neuron evaluates coalitions based on the presence of preferred partners. To capture this behavior, we model the setting as a *top-responsive game*. In a top-responsive game, every player $i$ associates each coalition $S \ni i$ with a unique *choice set* $ch(i, S) \subseteq S$ that represents the subset of partners most important to $i$. Preferences are then determined entirely by these choice sets: a player prefers one coalition over another if its choice set is ranked higher, and if two coalitions yield the same choice set, the smaller coalition is favored. This restriction makes coalition evaluation tractable, as each neuron only needs to consider its most valued partners rather than all possible groups.

The top responsive framework is flexible, as choice sets may consist of a single strong partner, multiple valued partners, or even subsets selected according to synergy between members. The key requirement is that choice sets are uniquely defined and utilities are represented in an *informative* way, so that distinct choice sets correspond to distinct utility "buckets". Under these conditions, the Top-Covering algorithm (Alcalde & Revilla, 2004; Dimitrov & Sung, 2007) can be applied to efficiently compute an $(\varepsilon, \delta)$ PAC-stable partition (Sliwinski & Zick, 2017). This enables us to identify groups of neurons that form stable coalitions under the distribution of observed samples. Further details and extensions are provided in Appendix A.1.

## 2.2 Transformer MLPs and Latent Feature Formation

Each LLM transformer block contains a gated MLP that expands the hidden state, applies a non-linearity, and then projects it back to the model dimension. Let the hidden vector entering the MLP at layer $\ell$ be $\vec{h} \in \mathbb{R}^{d_{\text{model}}}$, and let $d_{\text{ff}} > d_{\text{model}}$ denote the intermediate width. In LLaMA-3-style architectures (Dubey et al., 2024), the computation proceeds as:

$$\vec{z}_{\text{up}} = W_{\text{up}}\vec{h}, \quad \vec{z}_{\text{gate}} = W_{\text{gate}}\vec{h}; \quad \vec{g} = \text{SiLU}(\vec{z}_{\text{gate}}) \odot \vec{z}_{\text{up}}, \quad \vec{h}' = W_{\text{down}}\vec{g}.$$

where $W_{\text{up}}, W_{\text{gate}} \in \mathbb{R}^{d_{\text{ff}} \times d_{\text{model}}}$ and $W_{\text{down}} \in \mathbb{R}^{d_{\text{model}} \times d_{\text{ff}}}$. The element-wise product $\vec{g}$ binds the *gate* signal – which selects or suppresses coarse abstractions – with the *up* signal that carries candidate feature directions. $W_{\text{down}}$ then recombines these activated features. During this process, abstract features may *emerge* (via new activation directions), *merge* (when multiple features co-activate), *split* (when previously unified features diverge), or *disappear* (if suppressed by gating) (Elhage et al., 2021; Tian et al., 2023).

**LoRA-adapted projections.** In our setup, only the MLP projection matrices are fine-tuned using Low-Rank Adaptation (LoRA) (Hu et al., 2022). For any weight matrix $W \in \mathbb{R}^{m \times n}$, LoRA introduces a low-rank update of the form:

$$\tilde{W} = W + \Delta W, \quad \Delta W = \frac{\alpha}{r}AB^{\top}, \tag{1}$$

where $A \in \mathbb{R}^{m \times r}$ and $B \in \mathbb{R}^{n \times r}$ are the learned parameters, $r$ is the rank, and $\alpha$ is a scaling factor. When applied to $W_{\text{up}}$ and $W_{\text{gate}}$, we obtain:

$$\vec{z}_{\text{up}} = (W_{\text{up}} + \Delta W_{\text{up}})\vec{h}, \quad \vec{z}_{\text{gate}} = (W_{\text{gate}} + \Delta W_{\text{gate}})\vec{h}.$$

The updates $\Delta W_{\text{up}}$ and $\Delta W_{\text{gate}}$ are low-rank, as a result, they introduce only a small set of new feature directions in the high-dimensional MLP space. But because these directions are diffused across neurons, visual inspection of weight updates reveals no obvious structure—leading to our central question: *which subsets of neurons cooperate to encode task-specific behavior under LoRA?*

In Section 3, we develop a game-theoretic framework that directly identifies these functional coalitions, revealing how LoRA's parameter-efficient updates create localized but coordinated changes

that encode task-relevant abstractions without requiring exhaustive analysis of all possible neuron combinations.

## 3 METHODOLOGY

We present a game-theoretic framework to identify and track *latent coalitions*—cooperating groups of neurons within MLP layers of LoRA-tuned transformer models. Our approach consists of two stages: first, we formalize the intra-layer coalition discovery as a game with hedonic utilities and apply the PAC-Top-Cover algorithm to find stable neuron groupings; second, we connect these coalitions across layers using maximum-weight bipartite matching to trace how these abstract computational units evolve through consecutive layers in the network hierarchy.

### 3.1 PROBLEM STATEMENT: COALITION DISCOVERY AND TRACKING IN TRANSFORMER MLPS

Let $L$ be a transformer-based LLM fine-tuned for a scalar prediction task (e.g., relevance scoring) via LoRA. Let $\ell \in \{1, 2, \ldots, d\}$ denote an MLP layer in the network, where $d$ is the total number of layers, with $n = d_{\text{ff}}$ neurons in its intermediate dimension. Denote the down-projection weight matrix as $W_{\text{down}}^{(\ell)} \in \mathbb{R}^{d_{\text{model}} \times n}$, where each column $[W_{\text{down}}^{(\ell)}]_{\cdot,i}$ represents the learned projection vector for neuron $i$. Here neuron $i$, refers to the $i^{th}$ MLP channel in $d_{\text{ff}}$.

Our goal is to identify a partition $\pi^{(\ell)} = \{C_1, C_2, \ldots, C_k\}$ of neurons in layer $\ell$ such that each subset $C_i \subseteq \{1, \ldots, n\}$ captures a set of neurons that exhibit strong *synergy*—cooperative behavior in forming a semantic unit. We define synergy through a pairwise valuation function $\phi_{ij}$. Then, across layers, we aim to match coalitions from $\pi^{(\ell)}$ to those in $\pi^{(\ell+1)}$, enabling us to model feature *persistence*, *splitting*, *merging*, and other dynamic events.

### 3.2 CONSTRUCTING PAIRWISE VALUATIONS AND UTILITY SCORES

PAC-Top-Cover uses samples of coalitions $S \sim D$ to estimate each agent's top-k choice set within the remaining pool. We first compute pairwise valuations $\phi_{ij}$, which quantify affinity or synergy between neurons. We instantiate two complementary valuation functions:

**Orthogonal-Co-Activation (OCA).** This approach combines two intuitions: neurons with orthogonal weight vectors may capture complementary features, while neurons with high activation correlation may process similar patterns. For neuron pair $(i, j)$, we define:

$$\phi_{\text{OCA}}(i,j) = \left(1 - |\cos(W_i, W_j)|\right) \rho(a_i, a_j), \quad \rho(a_i, a_j) = \frac{\text{Cov}[a_i, a_j]}{\sigma_i \sigma_j}$$

where $W_i$ is the $i$-th column of $W_{\text{down}}^{(\ell)}$ (neuron $i$'s output weights), and $a_i$ denotes neuron $i$'s activations. The cosine term favors pairs with dissimilar weight vectors, while the correlation term captures their collaborative activation patterns (Pearson's correlation).

**Pairwise Ablation Synergy (PAS).** To directly measure the synergistic interaction between neurons $i$ and $j$, we compute the second-order interaction effect through ablation. Let $\ell(x)$ denote the model's logit output[2] for input $x$, and $\ell_{-S}(x)$ denote the logit when neurons in set $S$ are ablated (set to their pre-LoRA weight). The true interaction between neurons $i$ and $j$ is $\phi_{\text{PAS}}(i, j)$, where:

$$\Delta_{ij}(x) := \ell(x) - \ell^{-i}(x) - \ell^{-j}(x) + \ell^{-(i,j)}(x)$$

$$\phi^{\text{PAS}}(i,j) = \mathbb{E}_x[\Delta_{ij}(x)]$$

---

[2] We use a *layer-local logit* $\ell^{(\ell)}(x) = w^\top h'^{(\ell)}(x) + b$, i.e. the scalar score obtained *immediately after* the layer-$\ell$ MLP (including residual addition) but *before* entering block $\ell + 1$. Our goal is to discover coalitions that are intrinsically synergistic at the point they are formed. $w, b$ are cloned from the final task head and kept fixed for all layers. For readability we drop the superscript when the layer is clear from context.

so $\phi^{\text{PAS}}(i,j) > 0$ indicates that jointly ablating $(i,j)$ changes the layer-local score more than additively (synergy).

This measures how the joint ablation of both neurons differs from the sum of individual ablations. For computational efficiency with large $n$, we approximate this using second-order derivatives:

$$\phi^{\text{PAS}}(i,j) \approx \mathbb{E}_x\left[\frac{\partial^2 \ell(x)}{\partial a_i\, \partial a_j}\, \Delta a_i(x)\, \Delta a_j(x)\right], \qquad \Delta a_i(x) := a_i^{\text{LoRA}}(x) - a_i^{\text{base}}(x)$$

where the mixed partial derivative captures the interaction between neuron activations.

We experiment with both OCA (structural heuristic) and PAS (functional ablation-based) valuations to test robustness of our framework. In both the above defined pairwise valuation functions, positive values indicate *synergy* (neurons cooperate to produce information neither could alone), while negative values indicate *redundancy* (neurons provide overlapping information). We now use these valuation functions, to compute choice sets, which is used to compute the utility of a neuron in a set that is used by the PAC Top-Cover algorithm.

*Multi-Friend Choice Sets (MFC).* In this formation, each neuron is allowed to anchor its preference not on a single partner but on a *set of top-$k$ partners*. For player $i$, the choice set within coalition $S$ is

$$Ch(i,S) = \arg \max_{\substack{T \subseteq S\setminus\{i\} \\ |T|=k}} \sum_{j\in T} \phi_{ij},$$

with ties broken deterministically to ensure uniqueness. Utilities are then defined as $u_i(S) = \frac{1}{k}\sum_{j\in Ch(i,S)} \phi_{ij}$. This normalized model captures *multi-partner synergy*, where a neuron's activation is meaningful only when several complementary features are present. We refer to this algorithmic instantiation as **Hedonic-MFC**.

### 3.3 THE PAC TOP-COVER ALGORITHM.

We compute a PAC-stable partition using the PAC-Top-Cover procedure (Sliwinski & Zick, 2017), instantiated with our MFC utilities from §3.2. All PAC-stability statements in this section are with respect to a distribution $D$ over candidate coalitions $S \subseteq N$ induced by a coalition sampler.

*Coalition sampling distribution $D$.* In practice, we implement $D$ by repeatedly drawing a subset size $s \sim \text{Unif}\{k_{\min}, \ldots, k_{\max}\}$ and then sampling a coalition $S \subseteq N$ uniformly at random among subsets of size $s$.[3] We draw $m$ i.i.d. samples $S^{(1)}, \ldots, S^{(m)} \sim D$ per round.

*Estimating best sampled coalitions and choice sets.* For each player $i \in N$, we retain the best sampled coalition containing $i$ under the MFC utility:

$$T_i^\star \in \arg \max_{t\in[m]\,:\,i\in S^{(t)}} u_i\big(S^{(t)}\big), \tag{2}$$

breaking ties deterministically. We then estimate $i$'s (top-$k$) choice set from $T_i^\star$ as

$$B_i := \text{TOPK}_i(T_i^\star) \subseteq T_i^\star \setminus \{i\}, \tag{3}$$

where $\text{TOPK}_i(T)$ returns the $k$ elements of $T \setminus \{i\}$ with the largest $\phi_{ij}$ values (or all available elements if $|T \setminus \{i\}| < k$).

*Preference digraph and closed sink components.* We construct a directed graph $G = (N, E)$ with edges

$$(i,j) \in E \iff j \in B_i. \tag{4}$$

Intuitively, $i$ points to the partners it most prefers among its best sampled coalition. PAC-Top-Cover outputs coalitions corresponding to sink strongly-connected components (SCCs) that are closed under these estimated choice sets. Concretely, we compute the SCC decomposition of $G$ and select sink SCCs $C$ satisfying the closure condition

$$\forall i \in C, \;\; B_i \subseteq C. \tag{5}$$

---

[3]Other proposal distributions can be substituted without changing the algorithm below; $D$ is simply the induced distribution of the chosen sampler.

Each such set $C$ is emitted as a coalition.

*Iterative removal.* After outputting a coalition $C$, we remove its vertices from the remaining set of players and repeat the above procedure on the residual set until all players are assigned to a coalition.

The algorithm is detailed in Appendix C. With $m = \text{poly}\big(n, 1/\varepsilon, \log(1/\delta)\big)$ i.i.d. samples per round from $D$, PAC-Top-Cover returns a partition that is $\varepsilon$-approximately stable with probability at least $1 - \delta$ (with respect to $D$). This provides theoretical support that the discovered coalitions reflect robust cooperative structure among neurons under the sampling distribution. We next ask how the coalitions identified at one layer relate to those in subsequent layers.

## 3.4 Tracking Coalitions Across Layers

Our hypothesis is that coalitions capture intermediate features that may *persist*, *merge*, *split*, or *disappear* as computation proceeds through the network. Tracking these transitions provides an exploratory view of how features evolve with depth.

For each pair of coalitions $(C, C')$ from consecutive layers $\ell$ and $\ell + 1$, we measure their *interaction mass*, a heuristic that quantifies how strongly one coalition influences the next:

$$M(C, C') \;=\; \frac{1}{|C| \cdot |C'|} \sum_{p \in C} \sum_{q \in C'} \left( \left| W_{\text{up}}^{(\ell+1)}[q, p] \right| + \left| W_{\text{gate}}^{(\ell+1)}[q, p] \right| \right) \cdot A_p,$$

where $W_{\text{up}}^{(\ell+1)}, W_{\text{gate}}^{(\ell+1)} \in \mathbb{R}^{d_{\text{ff}} \times d_{\text{model}}}$ are the LoRA-adapted projection matrices of layer $\ell + 1$, $q \in \{1, \ldots, d_{\text{ff}}\}$ indexes target neurons in layer $\ell+1$, and $p \in \{1, \ldots, d_{\text{model}}\}$ indexes residual-stream dimensions feeding into layer $\ell+1$.[4] This formulation captures both the additive ($W_{\text{up}}$) and multiplicative gating ($W_{\text{gate}} \times \text{SiLU}$) pathways, while normalizing by coalition sizes ensures comparability across widths.

We assemble the interaction masses into a bipartite matrix and solve a maximum-weight matching problem to align coalitions across layers. For each match, we compute the fraction of a source coalition's output that flows into a target ($\alpha$) and the fraction of a target's input originating from that source ($\beta$). These ratios allow us to classify transitions into persistence (both high), splitting (low $\alpha$, high $\beta$), merging (high $\alpha$, low $\beta$), or disappearance (both low).

We stress that this analysis is exploratory. Transformers have residual connections, so neurons at layer $\ell$ influence all deeper layers, not just $\ell + 1$. Our method only captures local dynamics and likely underestimates long-range interactions, but it offers a first step toward visualizing how abstract feature groups may evolve through the network.

## 4 Experiments

We empirically validate our framework on three LLM architectures and three scalar-output IR tasks. We first describe models, tasks, and baselines, then present evaluation protocols and results (Tables 1, 2, 3, Appendix 6).

**Models.** We study LLaMA-3.1-8B (Dubey et al., 2024), Mistral-7B-v0.1 (Jiang et al., 2023), and Pythia-6.9B (Biderman et al., 2023), each adapted via LoRA (rank $r = 8$) restricted to MLP layers 7–14. Preliminary analysis showed these layers carry the strongest task-specific activity (Nijasure et al., 2025). Fine-tuning uses AdamW ($\eta = 2 \times 10^{-4}$, batch size 128, 3 epochs), with all base weights frozen. Performance of these fine-tuned LoRA models is further documented in Appendix F.

**Tasks.** Tasks are scalar objectives defined over query–document pairs from MS MARCO (Bajaj et al., 2018): (1) Covered-Query-Term Ratio (*CQTR*) = fraction of query terms present in the document, (2) Mean of Stream-Length Normalized Term Frequency (*Mean-TF/L*) = mean of length-normalized term frequencies, (3) Relevance Modelling (*RM*) = supervised passage ranking. CQTR

---

[4]In practice, we aggregate each source coalition $C$'s outgoing contribution onto the residual stream before measuring its influence on targets in layer $\ell+1$; $A_p = \mathbb{E}_x[|h_p^{(\ell)}(x)|]$ denotes the mean absolute residual activation on dimension $p$ over the training distribution.

and Mean-TF/L use MSE loss, RM uses NDCG. Models are trained on 500k pairs, validated on 5k, and evaluated OOD on TREC DL-19/20.

**Baselines.** We compare five coalition builders: *Random* (uniform neuron subsets with matched size histogram), *K-means (Spherical)* (on $\ell_2$-normalized mean activations, $k$ matched to Hedonic partition), *Hierarchical (Ward+cos)* (agglomerative under cosine distance, cut at same $k$), *Hedonic-OCA* (PAC-Top-Cover with $\phi_{\text{OCA}}$), *Hedonic-PAS* (PAC-Top-Cover with $\phi_{\text{PAS}}$).

For Hedonic sampling we draw $m = 8\times10^5$ candidate coalitions (size $[2, 10]$), retain top $\omega = 8\times10^4$ by utility, and use $\varepsilon = \delta = 0.1$. Choice sets use top-3 partners. Cross-layer matching uses thresholds $(\alpha_{\text{hi}}, \alpha_{\text{lo}}) = (0.7, 0.1)$ tuned on a 1% held-out split. All methods run on 4×A100-80GB GPUs; PAC-Top-Cover completes in 90 min (OCA) and 280 min (PAS). All numbers are averaged over 3 seeds with 95% confidence intervals.

**Evaluation.** We first report intrinsic synergy metrics (Appendix G) as diagnostics, then evaluate coalitions extrinsically with three tests:

- *OOD Drop.* For coalition $C$, we measure the performance drop on $\mathcal{D}_{\text{OOD}}$ (DL-19/20) when $C$ is ablated (neurons reset to pre-LoRA weights):

$$\Delta\mathcal{M}(C) = \mathcal{M}(\{\ell(x)\}) - \mathcal{M}(\{\ell_{-C}(x)\}),$$

  where $\mathcal{M}$ is NDCG@10 for RM and $-$MSE for CQTR/Mean-TF/L. Larger $\Delta\mathcal{M}(C)$ indicates greater functional importance.

- *Feature Alignment.* Each coalition's mean activation $a_C(x)$ is compared with known IR heuristics (list of MSLR features Qin & Liu (2013b)). Alignment is defined as the maximum squared Pearson correlation:

$$R^2(C) = \max_j \text{ Corr}^2(a_C(x), f_j(x)).$$

- *Coalition Predictivity.* Coalitions are treated as macro-features $A(x) \in \mathbb{R}^k$. A ridge regression $\hat{y}(x) = w^\top A(x)$ is trained on MS MARCO and evaluated OOD; we report $R^2$ for RM, CQTR, and Mean-TF/L.

Next, we discuss the results reported in Tables 1 (extrinsic coalition evaluation), Table 2(coalition predictivity), Table 3 (coalition transfer dynamics) and Appendix Table 6 (intrinsic coalition evaluation).

**Experimental Results.**

**Functional importance and interpretability (Table 1).** Across all three models and tasks, hedonic coalitions are markedly more *causal* and *interpretable* than clustering or random partitions. Ablating a single hedonic coalition (ablation = restoring those neurons to their pre-LoRA state) yields the largest OOD performance drops: for CQTR on LLaMA/Mistral/Pythia the OOD drop rises from $\approx$0.02–0.03 (K-means/Hier.) to 0.11–0.14 with Hedonic-PAS—about a **3–5×** increase; similar gaps hold for Mean-TF/L (0.10–0.13 vs. 0.02–0.03) and RM (0.14–0.17 vs. 0.03–0.04). At the same time, coalition activations align far more strongly with IR heuristics: alignment $R^2$ climbs from $\sim$0.11–0.18 (clustering) to **0.55–0.67** (Hedonic-PAS), with Hedonic-OCA consistently second-best ($\approx$0.38–0.49). Confidence intervals are narrow throughout, indicating stable estimates over seeds. Taken together, these results show that hedonic coalitions are both **functionally indispensable**—their removal produces large OOD degradation—and **semantically grounded**, tracking BM25/IDF/coverage signals far better than baselines.

**Predictive macro-features (Table 2).** Treating each coalition as a macro-feature and training a ridge regressor on MS MARCO, we see large generalization gains on DL-19/20. Averaged over LLaMA, Mistral, and Pythia, Hedonic-PAS attains $R^2 =$ **0.43/0.42/0.47** on CQTR/Mean-TF/L/RM, roughly **2–3×** higher than K-means/Hier. ($\approx$0.15–0.21) and far above Random ($\approx$0.08–0.12). Hedonic-OCA also performs strongly ($\approx$0.33–0.38), reinforcing the pattern from the extrinsic ablations: utilities that respect *synergy* (PAS) or *partner preference* (OCA) produce coalitions that behave like **robust, transferable features**, not just co-activation clusters. This bridges intrinsic synergy to downstream utility: coalitions that score high on synergy also yield higher OOD predictivity.

Table 1: Extrinsic Evaluation: OOD Drop ($\uparrow$) and Feature Alignment $R^2$ ($\uparrow$) on DL-19/20. Mean $\pm 95\%$ CI across three seeds. Larger values indicate more functionally important and interpretable coalitions.

| Task / Algorithm | LLaMA-3.1 | | Mistral | | Pythia | |
|---|---|---|---|---|---|---|
| | OOD Drop | Align $R^2$ | OOD Drop | Align $R^2$ | OOD Drop | Align $R^2$ |
| **Covered Query Term Ratio** | | | | | | |
| Random | $0.01\pm0.01$ | $0.05\pm0.02$ | $0.00\pm0.01$ | $0.06\pm0.02$ | $0.01\pm0.01$ | $0.05\pm0.02$ |
| K-means | $0.02\pm0.01$ | $0.12\pm0.02$ | $0.03\pm0.01$ | $0.13\pm0.02$ | $0.02\pm0.01$ | $0.11\pm0.02$ |
| Hier. clustering | $0.03\pm0.01$ | $0.15\pm0.02$ | $0.03\pm0.01$ | $0.16\pm0.02$ | $0.03\pm0.01$ | $0.14\pm0.02$ |
| Hedonic (OCA) | $0.07\pm0.01$ | $0.41\pm0.02$ | $0.09\pm0.01$ | $0.44\pm0.02$ | $0.08\pm0.01$ | $0.45\pm0.02$ |
| Hedonic (PAS) | $\mathbf{0.11}\pm0.005$ | $\mathbf{0.58}\pm0.01$ | $\mathbf{0.13}\pm0.006$ | $\mathbf{0.61}\pm0.01$ | $\mathbf{0.14}\pm0.006$ | $\mathbf{0.63}\pm0.01$ |
| **Mean of Normalized Term Frequency** | | | | | | |
| Random | $0.01\pm0.01$ | $0.04\pm0.02$ | $0.01\pm0.01$ | $0.05\pm0.02$ | $0.01\pm0.01$ | $0.05\pm0.02$ |
| K-means | $0.02\pm0.01$ | $0.11\pm0.02$ | $0.02\pm0.01$ | $0.12\pm0.02$ | $0.02\pm0.01$ | $0.10\pm0.02$ |
| Hier. clustering | $0.03\pm0.01$ | $0.14\pm0.02$ | $0.03\pm0.01$ | $0.15\pm0.02$ | $0.03\pm0.01$ | $0.13\pm0.02$ |
| Hedonic (OCA) | $0.06\pm0.01$ | $0.38\pm0.02$ | $0.08\pm0.01$ | $0.41\pm0.02$ | $0.07\pm0.01$ | $0.40\pm0.02$ |
| Hedonic (PAS) | $\mathbf{0.10}\pm0.005$ | $\mathbf{0.55}\pm0.01$ | $\mathbf{0.12}\pm0.006$ | $\mathbf{0.59}\pm0.01$ | $\mathbf{0.13}\pm0.006$ | $\mathbf{0.60}\pm0.01$ |
| **Relevance Modelling** | | | | | | |
| Random | $0.02\pm0.01$ | $0.06\pm0.02$ | $0.01\pm0.01$ | $0.07\pm0.02$ | $0.02\pm0.01$ | $0.06\pm0.02$ |
| K-means | $0.03\pm0.01$ | $0.13\pm0.02$ | $0.04\pm0.01$ | $0.14\pm0.02$ | $0.03\pm0.01$ | $0.13\pm0.02$ |
| Hier. clustering | $0.04\pm0.01$ | $0.17\pm0.02$ | $0.04\pm0.01$ | $0.18\pm0.02$ | $0.04\pm0.01$ | $0.16\pm0.02$ |
| Hedonic (OCA) | $0.09\pm0.01$ | $0.47\pm0.02$ | $0.10\pm0.01$ | $0.49\pm0.02$ | $0.09\pm0.01$ | $0.48\pm0.02$ |
| Hedonic (PAS) | $\mathbf{0.14}\pm0.006$ | $\mathbf{0.63}\pm0.01$ | $\mathbf{0.16}\pm0.006$ | $\mathbf{0.65}\pm0.01$ | $\mathbf{0.17}\pm0.007$ | $\mathbf{0.67}\pm0.01$ |

Table 2: Coalition Predictivity ($R^2$ on OOD sets DL-19/20), averaged across three LLMs (LLaMA-3.1, Mistral, Pythia). Coalitions are used as macro-features in ridge regression trained on MS MARCO. Hedonic coalitions yield substantially higher $R^2$ than clustering or random baselines.

| Algorithm | CQTR | Mean-TF/L | Relevance (RM) |
|---|---|---|---|
| Random | $0.08\pm0.02$ | $0.09\pm0.02$ | $0.12\pm0.02$ |
| K-means | $0.16\pm0.01$ | $0.15\pm0.01$ | $0.21\pm0.01$ |
| Hier. clustering | $0.18\pm0.01$ | $0.17\pm0.01$ | $0.21\pm0.01$ |
| Hedonic (OCA) | $0.34\pm0.01$ | $0.33\pm0.01$ | $0.38\pm0.01$ |
| Hedonic (PAS) | $\mathbf{0.43}\pm0.008$ | $\mathbf{0.42}\pm0.008$ | $\mathbf{0.47}\pm0.008$ |

**Coalition dynamics across depth (Table 3).** Across layers $7\rightarrow14$, three trends are consistent: (i) *vanish dominates* (typically *60–75%* of coalitions disappear at the next layer), indicating downstream MLPs act as *filters/refiners* rather than combiners; (ii) *splits are common* ($\approx$20–50%, depending on task/layer), suggesting feature *refinement* is more prevalent than wholesale reuse; and (iii) *merges are near-zero*, implying whole motifs are rarely recomposed from separate groups. Persistence is generally low ($<\sim12\%$), with a mild *delayed persistence uptick* around $12\rightarrow13$ for CQTR and RM ($\approx$8–11%), echoing a "late stabilization" phase. Mean-TF/L exhibits the strongest pruning (vanish $>70\%$ across several transitions), consistent with simple frequency statistics being isolated early and aggressively culled later. These dynamics support our central claim: **cooperative units are formed, then predominantly pruned or refined rather than fused**, aligning with the heavy-tailed coalition sizes and the functional importance patterns observed above.

## 5 DISCUSSION

*SAEs vs Hedonic Neurons.* Sparse Autoencoders (SAEs) (Huben et al., 2024) uncover interpretable features by learning sparse dictionaries that reconstruct activations and disentangle polysemantic units. In contrast, our framework keeps neurons as primitives and asks how they cooperate. By modeling them as agents in a hedonic game, we capture nonlinear synergies: coalitions whose joint ablation impacts behavior beyond the sum of parts. Unlike SAEs, which re-express activation space, hedonic coalitions are grounded in weight geometry and preference structure, surfacing cooperative

Table 3: Dynamics of coalitions across layers 7–14 for three tasks. Each cell shows percentage of coalitions exhibiting the event relative to all coalitions present in the *source* layer (except *merge*).

| Layer → | Mistral | | | | LLaMA | | | | Pythia | | | |
|---|---|---|---|---|---|---|---|---|---|---|---|---|
| | Persist | Merge | Split | Vanish | Persist | Merge | Split | Vanish | Persist | Merge | Split | Vanish |
| **Covered Query Term Ratio** | | | | | | | | | | | | |
| 7 → 8 | 12.1% | 0.0% | 28.9% | 59.0% | 3.2% | 0.0% | 35.4% | 61.4% | 7.8% | 0.0% | 31.9% | 60.3% |
| 8 → 9 | 4.8% | 0.0% | 38.4% | 56.8% | 5.1% | 0.0% | 28.6% | 66.3% | 4.9% | 0.0% | 33.7% | 61.4% |
| 9 → 10 | 6.2% | 0.0% | 31.5% | 62.3% | 4.8% | 0.0% | 30.2% | 65.0% | 5.5% | 0.0% | 30.8% | 63.7% |
| 10 → 11 | 3.8% | 0.0% | 29.7% | 66.5% | 7.9% | 0.0% | 32.1% | 60.0% | 5.9% | 0.0% | 30.9% | 63.2% |
| 11 → 12 | 4.2% | 0.0% | 27.8% | 68.0% | 3.5% | 0.0% | 29.8% | 66.7% | 3.9% | 0.0% | 28.8% | 67.3% |
| 12 → 13 | 11.3% | 0.0% | 30.2% | 58.5% | 6.8% | 0.0% | 27.4% | 65.8% | 9.1% | 0.0% | 28.8% | 62.1% |
| 13 → 14 | 10.5% | 0.0% | 24.3% | 65.2% | 7.2% | 0.0% | 23.1% | 69.7% | 8.9% | 0.0% | 23.7% | 67.4% |
| **Stream Length Normalized Term Frequency** | | | | | | | | | | | | |
| 7 → 8 | 6.4% | 0.5% | 35.8% | 57.3% | 2.1% | 0.0% | 19.7% | 78.2% | 4.3% | 0.3% | 28.4% | 67.0% |
| 8 → 9 | 1.8% | 0.2% | 51.2% | 46.8% | 3.8% | 0.1% | 20.3% | 75.8% | 2.8% | 0.1% | 36.2% | 60.9% |
| 9 → 10 | 2.9% | 0.1% | 23.1% | 73.9% | 3.4% | 0.0% | 22.8% | 73.8% | 3.2% | 0.0% | 22.9% | 73.9% |
| 10 → 11 | 1.3% | 0.3% | 23.7% | 74.7% | 5.9% | 0.2% | 25.5% | 68.4% | 3.6% | 0.2% | 24.6% | 71.6% |
| 11 → 12 | 1.2% | 0.1% | 22.9% | 75.8% | 1.7% | 0.0% | 24.1% | 74.2% | 1.4% | 0.0% | 23.5% | 75.1% |
| 12 → 13 | 6.3% | 0.4% | 37.8% | 55.5% | 3.2% | 0.1% | 19.8% | 76.9% | 4.8% | 0.2% | 29.3% | 65.7% |
| 13 → 14 | 7.1% | 0.2% | 19.5% | 73.2% | 4.7% | 0.0% | 17.1% | 78.2% | 5.9% | 0.1% | 18.3% | 75.7% |
| **Relevance** | | | | | | | | | | | | |
| 7 → 8 | 8.2% | 0.0% | 32.7% | 59.2% | 1.5% | 0.0% | 22.4% | 76.1% | 5.2% | 0.0% | 28.1% | 66.7% |
| 8 → 9 | 2.1% | 0.0% | 46.8% | 51.1% | 3.5% | 0.0% | 22.8% | 73.7% | 2.8% | 0.0% | 35.3% | 61.9% |
| 9 → 10 | 3.5% | 0.0% | 26.3% | 70.2% | 3.9% | 0.0% | 25.5% | 70.6% | 3.7% | 0.0% | 25.9% | 70.4% |
| 10 → 11 | 2.0% | 0.0% | 26.5% | 71.4% | 6.7% | 0.0% | 28.3% | 65.0% | 4.2% | 0.0% | 27.4% | 68.4% |
| 11 → 12 | 1.7% | 0.0% | 25.4% | 72.9% | 2.0% | 0.0% | 26.5% | 71.4% | 1.9% | 0.0% | 25.9% | 72.2% |
| 12 → 13 | 8.0% | 0.0% | 34.0% | 58.0% | 4.0% | 0.0% | 22.0% | 74.0% | 6.1% | 0.0% | 28.2% | 65.7% |
| 13 → 14 | 8.7% | 0.0% | 21.7% | 69.6% | 5.4% | 0.0% | 18.9% | 75.7% | 7.1% | 0.0% | 20.3% | 72.6% |

"wiring-level" units already encoded in the parameters. The two approaches are complementary: SAEs expose monosemantic features, while hedonic analysis highlights how neurons collaborate to realize them.

*Coalition size distribution.* Coalition sizes follow a heavy-tailed Zipfian law: each layer contains a few large "macro" groups, mid-sized units, and many size-2 specialists, resembling vocabulary statistics in language. Disappearance rates rise after layer 12, suggesting deeper MLP blocks act more as feature filters than creators. Together, these findings imply that hedonic coalitions are natural computational units shaped by training dynamics—early layers construct rich representations, while later ones selectively retain task-relevant features.

# 6 RELATED WORK

Mechanistic interpretability of transformer LLMs has focused on understanding both individual neurons and structured groups. Geva et al. (2021) showed that feed-forward layers act as key–value memories, with neurons detecting input patterns (keys) and injecting values into the representation. Dai et al. (2022) identified "knowledge neurons" in MLPs that encode factual associations, demonstrating that small groups of neurons can robustly store discrete knowledge.

Beyond single-neuron analysis, Bricken et al. (2023) applied dictionary learning to extract sparse, interpretable features from polysemantic activations. Balagansky et al. (2025) tracked feature persistence and merging across layers, complementing our coalition-evolution view. Sparse probing (Gurnee et al., 2023) further revealed that early layers are highly polysemantic while deeper layers specialize, underscoring the need to model neuron groups and their dynamics. Weight-based methods also contribute: Davies (2025) decoded neuron weights into semantic concepts, while Pearce et al. (2024) and Bushnaq et al. (2025) developed direct weight-space feature discovery.

While hedonic games have rarely been explored in interpretability, Koulali and Koulali (Koulali & Koulali, 2023) showed their utility for feature selection, providing theoretical foundations for our approach. Our work extends these lines by explicitly framing neuron collaboration as a hedonic game,

enabling principled discovery and tracking of *stable coalitions* that serve as latent computational units in transformer MLPs.

# 7 Conclusion, Limitations and Future Work

We introduced **Hedonic Neurons**, a game-theoretic framework that models neurons in transformer MLPs as players in a top-responsive hedonic game. Using the PAC-Top-Cover algorithm with correlation-based (OCA) or ablation-based (PAS) valuations, we identified stable coalitions that capture cooperative structure beyond what clustering can reveal. Across three LLM architectures and scalar IR tasks, hedonic coalitions achieve average improvements of +0.29 *Pairwise* and +0.49 *Ratio* synergy over the strongest baseline, while extrinsic evaluations show they are functionally indispensable: ablations yield 3–5× larger OOD performance drops, alignment with IR heuristics rises from ∼0.15 to 0.55–0.67, and predictive $R^2$ improves from ∼0.20 to 0.43–0.47. Coalition dynamics further reveal that most groups vanish or split across depth, with merges rare and persistence limited, supporting the view that MLPs act primarily as filters and refiners of features.

Our approach has limitations: utilities depend on layer-local logits and second-order ablations, omitting higher-order interactions and attention mechanisms, and the current formulation yields disjoint coalitions despite early-layer polysemy. Future work will extend to overlapping coalitions via fractional hedonic games, integrate attention heads for joint sub-module analysis, and design low-variance estimators to reduce $O(n^2)$ ablation costs. Coupling hedonic discovery with concept-activation vectors may also yield interpretable primitives aligned with human-understandable features. Taken together, HedonicNeurons provides a principled foundation for uncovering how cooperative computational units emerge, evolve, and specialize in large-scale language models.

# 8 Reproducibility Statement

We provide all resources necessary to reproduce our experiments. We make our fine-tuned reranker checkpoints for Pythia, Mistral, and LLaMA3 models available on HuggingFace. The training dataset (Tevatron MSMARCO Passage Augmented) and evaluation dataset (TREC DL 2019) are publicly available, with preprocessing steps following the Tevatron MSMARCO implementation. All scripts used for coalition generation, partitioning, clustering baselines, and evaluation are included in the repository, along with deepspeed configuration files for finetuning. Coalition files (`.pkl`) and visualization outputs (Sankey plots) are also provided.

## Acknowledgments

This work was supported in part by the Center for Intelligent Information Retrieval, University of Massachusetts Amherst. Any opinions, findings and conclusions or recommendations expressed in this material are those of the authors and do not necessarily reflect those of the sponsor.

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

# A  HEDONIC GAMES PRELIMINARIES AND PAC TOP COVER INTUITION

## A.1  HEDONIC GAMES

A *hedonic game* (Dreze & Greenberg, 1980) is defined by a finite set of players $N = \{1, \ldots, n\}$ and, for each player $i$, a complete and transitive preference relation $\succ_i$ over the set $\mathcal{N}_i = \{S \subseteq N \mid i \in S\}$ of coalitions that contain $i$. A *coalition structure* (or *partition*) is a set $\pi = \{C_1, \ldots, C_k\}$ of disjoint non-empty coalitions whose union equals $N$. Throughout this appendix we assume that preferences are given by real-valued utilities $v_i : \mathcal{N}_i \to \mathbb{R}$ so that $S \succ_i T \Leftrightarrow v_i(S) > v_i(T)$.[5]

## A.2  CORE STABILITY

Given a partition $\pi$ and a coalition $S \subseteq N$, we say that $S$ *blocks* $\pi$ if every $i \in S$ strictly prefers $S$ to her coalition in $\pi$, i.e. $S \succ_i \pi(i)$. A partition is *core-stable* (or simply *in the core*) if it is not blocked by any coalition. Core stability captures the idea that no subset of players has a joint incentive to deviate.

## A.3  WHY FULL PREFERENCE LEARNING IS INFEASIBLE

Precisely learning all utilities $v_i(S)$ is unrealistic because the number of coalitions grows exponentially ($|\mathcal{N}_i| = 2^{n-1}$). Even if we could query any coalition, the sample complexity implied by the pseudo-dimension of general hedonic games is super-polynomial (Proposition 4.9 in (Sliwinski & Zick, 2017)). Hence, any practical method must settle for *approximate* stability based on samples rather than complete preference elicitation.

## A.4  PAC-LEARNING FRAMEWORK FOR HEDONIC GAMES

Following (Sliwinski & Zick, 2017), let $D$ be an unknown but fixed distribution over coalitions. A partition $\pi$ is $\varepsilon$-*PAC stable under* $D$ if

$$\Pr_{S \sim D}\big[S \text{ blocks } \pi\big] \; < \; \varepsilon.$$

An algorithm A *PAC-stabilises* a class $\mathcal{H}$ of hedonic games if, for any game $G \in \mathcal{H}$, distribution $D$, and parameters $(\varepsilon, \delta)$, A outputs—with probability at least $1 - \delta$—an $\varepsilon$-PAC-stable partition using a number of samples polynomial in $(n, 1/\varepsilon, \log(1/\delta))$.

## A.5  INTUITION BEHIND THE TOP-COVER ALGORITHM

Under *top responsive* utilities ($v_i(S) = \sum_{j \in S \setminus \{i\}} u_{ij}$), players exhibit *top-responsiveness*: their evaluation of a coalition is determined by the "best" members plus a size penalty (Alcalde & Revilla, 2004). TOP-COVER exploits this property iteratively:

  (i) using samples, approximate each player's most preferred subset within the current residual set,

 (ii) build directed edges from each player to the members of that subset,

(iii) extract a strongly connected component of minimal size, form it as a coalition, and remove it,

 (iv) repeat until all players are assigned.

Each extracted coalition is unlikely to be blocked because every member already sees its best attainable partners within it with high probability.

## A.6  APPLICATION OF HEDONIC GAMES TO NEURAL NETWORKS.

Neurons in a transformer predominantly interact with a limited set of peers—those with highly correlated activations or complementary weights. Treating neurons as players whose utilities are

---

[5]See Section 2 of (Sliwinski & Zick, 2017) for an extensive discussion of numeric versus ordinal representations.

derived from such local synergies fits the additive model naturally. Sampling mini-batches of logits/activations supplies the coalitions needed by the PAC framework, letting us recover *approximately core-stable neuron groups* without exhaustively testing all neuron subsets.

## B  MAKING ADDITIVE UTILITIES TOP-RESPONSIVE

**Notation recap.**  For each ordered pair of distinct neurons $(i, j)$ we have a *pairwise synergy score* $\phi_{ij} \in \mathbb{R}$ (either $\phi_{\text{OCA}}$ or $\phi_{\text{PAS}}$; see §3.2). Write $\Phi$ for the $n \times n$ matrix with zeros on the diagonal.

### B.1  FROM ADDITIVE SCORES TO TOP-RESPONSIVE PREFERENCES

**Max-partner utility** : Fix a global parameter $k \geq 1$. For a coalition $S \subseteq N$ that contains player $i$ let

$$\text{Top}_k(i, S) \;=\; \underset{\substack{T \subseteq S \setminus \{i\} \\ |T| \leq k}}{\arg \max} \; \sum_{j \in T} \phi_{ij}.$$

We define

$$u_i(S) \;=\; \sum_{j \in \text{Top}_k(i,S)} \phi_{ij}, \quad \text{and} \quad \mathcal{C}_i(S) \;=\; \text{Top}_k(i, S).$$

When $k = 1$ this reduces to the familiar "best-friend" utility $u_i(S) = \max_{j \in S \setminus \{i\}} \phi_{ij}$.

**Lemma B.1** (Top-responsiveness).  *For every player $i$ the preference relation $\succeq_i$ induced by $u_i$ is* top-responsive*: for any two coalitions $S, T$ that contain $i$*

$$\mathcal{C}_i(S) \succ_i \mathcal{C}_i(T) \implies S \succ_i T.$$

*Proof.*  Let $S, T$ contain $i$ and assume $\mathcal{C}_i(S) \succ_i \mathcal{C}_i(T)$, i.e. $u_i(\mathcal{C}_i(S)) > u_i(\mathcal{C}_i(T))$. Because $u_i$ is *monotone* in the sense that enlarging a set never decreases its utility,[6] we have $u_i(S) \geq u_i(\mathcal{C}_i(S))$ and $u_i(T) = u_i(\mathcal{C}_i(T))$. Hence $u_i(S) > u_i(T)$, so $S \succ_i T$. □

**Lemma B.2** (Informative representation).  *Given the matrix $\Phi$ one can compute $\mathcal{C}_i(S)$ (and therefore the induced ranking) in $O(k\,|S|)$ time. Hence the utility representation is* informative *in the sense of Sliwinski and Zick(Sliwinski & Zick, 2017).*

*Proof.*  $\text{Top}_k(i, S)$ requires sorting at most $|S| - 1$ real numbers $\{\phi_{ij}\}_{j \in S \setminus \{i\}}$; the $k$ largest can be found in the stated time using a partial-selection routine. □

**Theorem B.3** (Applicability of PAC-Top-Cover).  *With utilities $u_i$ from Definition B.1 the induced hedonic game is top-responsive and informative. Consequently, Algorithm 1 outputs an $\varepsilon$-PAC-stable partition with probability $1 - \delta$ using $m = \text{poly}\big(n, \frac{1}{\varepsilon}, \log \frac{1}{\delta}\big)$ samples, exactly as in (Sliwinski & Zick, 2017).*

*Proof.*  Top-responsiveness follows from Lemma B.1; informativeness from Lemma B.2. The PAC-stability guarantee is therefore an immediate corollary of Theorem 3.4 in (Sliwinski & Zick, 2017). □

### B.2  COALITION-LEVEL VALUATION (FOR SAMPLING)

Algorithm 1 needs a scalar value for any sampled coalition $S$. We use the symmetric extension

$$\Phi(S) \;=\; \frac{1}{|S|} \sum_{i \in S} u_i(S) \;=\; \frac{1}{|S|} \sum_{i \in S} \sum_{j \in \text{Top}_k(i,S)} \phi_{ij}.$$

Intuitively, $\Phi(S)$ averages how strongly each member is bonded to its $k$ preferred partners within $S$. Plugging $\phi_{\text{OCA}}$ or $\phi_{\text{PAS}}$ in place of $\phi_{ij}$ yields the concrete scores used in our experiments. "Reservoir" sampling in line 4 of Algorithm 1 draws $m$ subsets $S$ with probability proportional to $\Phi(S)$, thereby prioritising high-synergy groups.

---

[6]Adding a partner can only increase the set of $k$ best partners or leave it unchanged.

---

**Algorithm 1** PAC Top-Cover for Top-$k$ Responsive Games (neurons)

---

**Require:** $\phi \in \mathbb{R}^{n \times n}$           $\triangleright$ pairwise affinity; $\phi_{ii} = 0$
       $k \in \mathbb{N}$               $\triangleright$ top-$k$ choice size
       $m, \omega$          $\triangleright$ reservoir size, per-round samples
       MINK, MAXK          $\triangleright$ sampled coalition sizes
       $(\varepsilon, \delta)$           $\triangleright$ PAC guidance for $m, \omega$

1: $R \leftarrow \{1, \ldots, n\}, \quad \pi \leftarrow \emptyset$
2: $S \leftarrow$ SAMPLECOALITIONS$(R, m, \text{MINK}, \text{MAXK})$          $\triangleright$ reservoir
    **Definition (top-$k$ utility and choice in a coalition).**
    For $i \in T$, let $P_i(T) = T \setminus \{i\}$. Let TOPK$_i(T)$ be the $k$ indices in $P_i(T)$
    with largest $\phi_{ij}$ (ties broken by smaller index); if $|P_i(T)| < k$, take all.
    Define $u_i^k(T) \triangleq \frac{1}{|\text{TOPK}_i(T)|} \sum_{j \in \text{TOPK}_i(T)} \phi_{ij}$.
3: **while** $R \neq \emptyset$ **do**
4:     $S_{\text{round}} \leftarrow$ first $\omega$ sets in $S$ that satisfy $T \subseteq R$; remove them from $S$
5:     **if** $|S_{\text{round}}| < \omega$ **then**          $\triangleright$ refresh if reservoir depleted
6:        $S \leftarrow S \cup$ SAMPLECOALITIONS$(R, m, \text{MINK}, \text{MAXK})$
7:     **end if**
8:     **for all** $i \in R$ **do**
9:        $\mathcal{T}_i \leftarrow \{T \in S_{\text{round}} : i \in T\}$
10:       **if** $\mathcal{T}_i = \emptyset$ **then**
11:         $B_i \leftarrow \{i\}$          $\triangleright$ degenerate self-loop
12:       **else**
13:         $T_i^\star \leftarrow \arg\max_{T \in \mathcal{T}_i} u_i^k(T)$ $\triangleright$ deterministic tie-break by $T$'s lexicographic index list
14:         $B_i \leftarrow$ TOPK$_i(T_i^\star)$          $\triangleright$ top-$k$ choice set of $i$ in $T_i^\star$
15:       **end if**
16:     **end for**
17:     Build digraph $G = (R, E)$ with edges $(i \rightarrow j)$ for all $j \in B_i$ (and optional $(i \rightarrow i)$ self-loops)
18:     Let $\mathcal{C} \leftarrow$ the set of sink strongly connected components of $G$
19:     **(closure check)** Keep only $X \in \mathcal{C}$ such that $\forall i \in X : B_i \subseteq X$
20:     Choose $X \in \mathcal{C}$ (e.g., smallest by size then lexicographic)     $\triangleright$ any sink closed SCC is valid
21:     $\pi \leftarrow \pi \cup \{X\}; \quad R \leftarrow R \setminus X$
22: **end while**
23: **return** $\pi$

---

# C   PAC TOP COVER ALGORITHM

## D  INFORMATION RETRIEVAL PRELIMINARIES

Information Retrieval (IR) involves retrieving documents that are likely to be relevant to a user's information need, typically represented as a query. A fundamental IR task is to return a ranked list of documents in descending order of (estimated) relevance. The quality of this ranking directly impacts the user experience in search engines, recommendation systems, and question answering applications.

### D.1  RELEVANCE MODEL

**Dense/Neural Re-ranker** is a language model (like RankLLaMa (Ma et al., 2024a)) which takes a query and text as input and produces a relevance score based on the similarity of the query to the provided text.

**Relevance Modeling vs Classification** Classification and relevance modeling are related but distinct approaches in information retrieval (IR). The term relevance model (Lavrenko & Croft, 2001) refers to a mechanism for estimating the likelihood of observing a particular word in documents that are relevant to a given information need or query, whereas classification assigns documents to predefined categories, such as relevant or non-relevant.

**Ranking Evaluation with NDCG**

In information retrieval, one commonly used metric to evaluate the effectiveness of ranking models is the Normalized Discounted Cumulative Gain (NDCG). NDCG assesses the quality of a ranked list by measuring the gain (or relevance) of documents based on their position in the list, giving higher weight to relevant documents that appear earlier. Formally, the Discounted Cumulative Gain (DCG) is computed as:

$$\text{DCG@k} = \sum_{i=1}^{k} \frac{2^{rel_i} - 1}{\log_2(i + 1)}$$

where $rel_i$ is the graded relevance of the document at position $i$. The NDCG is then computed by normalizing DCG by the ideal DCG (IDCG), which is the DCG for the optimal ranking:

$$\text{NDCG@k} = \frac{\text{DCG@k}}{\text{IDCG@k}}$$

NDCG scores range from 0 to 1, with 1 indicating a perfect ranking. In the DL19 dataset, each query-document pair is labeled with a relevance grade based on human annotations. These annotations are used to compute the NDCG score for a re-ranked list of documents, allowing us to quantify the effectiveness of our rerankers in retrieving the most relevant content at the top of the list.

In our work, we use **RankLLaMA**, a LLaMA-based reranking model trained to predict the relevance of a document given a query. The model takes as input a formatted string:

```
"query: {query}, passage: {passage}"
```

and outputs a score between 0 and 1, indicating the estimated relevance. We follow the training procedure described in the RankLLaMA paper (Ma et al., 2024a).

### D.2  COVERED QUERY TERM RATIO (CQTR)

**Covered Query Term Ratio (CQTR)** is a lexical feature that measures the proportion of unique query terms found in the document(Qin & Liu, 2013a). Formally:

$$\text{CQTR} = \frac{|\text{Query Terms} \cap \text{Document Terms}|}{|\text{Query Terms}|}$$

### D.3 MEAN TERM FREQUENCY PER DOCUMENT LENGTH (MTF/L)

**Mean Term Frequency per Document Length (MTF/L)** captures the average frequency of query terms normalized by the document length(Qin & Liu, 2013a). It is computed as:

$$\text{MTF/DL} = \frac{\sum_{t \in Q} \text{TF}_t(D)}{\text{Length}(D)}$$

To simplify interpretability tasks (by trying to restrict polysemanticity(Elhage et al., 2022)), we fine-tuned models on CQTR and MTF/L prediction tasks, with the same input structure as defined above. We do not claim that these two features are the most important for determining relevance; rather, they are easily understood signals that prior work has shown to be implicitly present in neural models (Chowdhury et al., 2025).

### D.4 DATASETS

**Datasets**:

- **MS MARCO**: A large-scale dataset consisting of real anonymized web search queries paired with relevant passages. It is a standard benchmark for training and evaluating re-ranking models. In our fine-tuning, we used a modified version of this dataset called MS MARCO Augmented (Tevatron, 2024) (Ma et al., 2025), which provides hard negatives from both CoconDenser(Gao et al., 2021) and BM25. [7]

- **DL-19 (TREC Deep Learning Track 2019)**: Contains high-quality relevance annotations for a subset of queries, commonly used for zero-shot and fine-tuned re-ranker evaluation. Craswell et al. provide more information and an overview of this dataset (Craswell et al., 2020).

---

[7]More details at https://microsoft.github.io/msmarco/

# E  FEATURE/LAYER CHOICE

Previous interpretability studies have been conducted on dense re-rankers, where Chowdhury et al. found that using linear probing, several traditional IR features show a high likelihood of being present in the forward pass activations of a dense re-ranker model(Chowdhury et al., 2025). Further behavioral analysis by Nijasure et al. observed that large language models (LLMs) tend to learn relevance-related features primarily in MLP layers 5 to 14 of re-ranker architectures(Nijasure et al., 2025).

Motivated by these insights, we focused our probing and editing experiments on this layer range (7–14) of Re-Ranker models. Figure 1 supports this choice: it shows $R^2$ scores for predicting MSLR features across all layers of the RankLLaMA-7b model using linear probing. Features like *covered query term number*, *covered query term ratio*, *mean of stream length normalized term frequency*, and *variance of $tf \cdot idf$* exhibit increasing prominence from the lower to mid layers. This trend might indicate that these layers are key to encoding relevance-related signals.

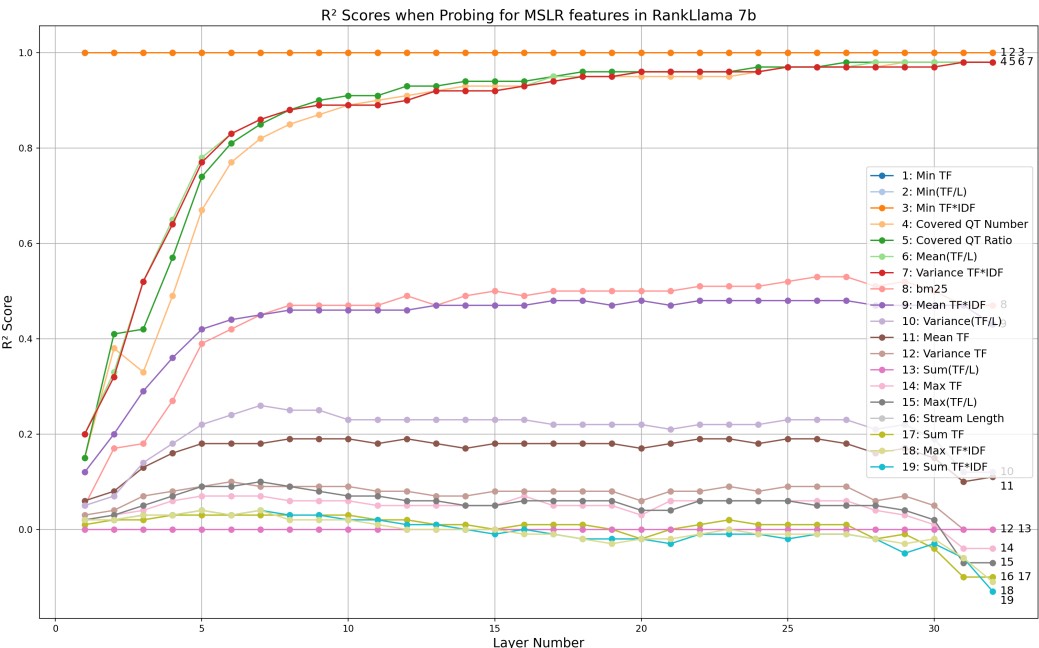

Figure 1: Probing for statistical features from the MSLR dataset in RankLlama2-7b model. Here $QT$ stands for Query Term, $TF$ stands for Term Frequency and $\cdot/L$ stands for length normalized. The graph lines indicate the presence of a particular feature along the layers of the LLM. Certain features like $Min\ TF * IDF$ show consistent presence across the layers. Other features like $Covered\ QT\ Number$, $Covered\ QT\ Ratio$, $Mean(TF/L)$ and $Variance\ TF * IDF$ show increasing prominence from the first layer to the last, ultimately playing an important role in making ranking decisions. Other MSLR features like $Sum(TF/L)$, $Max(TF/L)$, and $Sum\ TF * IDF$ show negative correlation with RankLlama decision making(Chowdhury et al., 2025).

# F  LLM Performance Evaluation

We used LoRA (rank 8) fine-tuning on MLP modules alone for all models described in this section. We had access to four A100 GPUs, depending on availability. We used DeepSpeed's Stage 0 configuration with the AdamW optimizer for fine-tuning all these models.

Two of the LLMs used in our experiments were fine-tuned on the MS MARCO dataset for 0.3 epochs using Mean Squared Error (MSE) as the loss function. The models were trained to predict statistical IR signals such as the Covered Query Term Ratio (CQTR) and the mean term frequency normalized by passage length (mean(TF/L)). Following this fine-tuning, the models were evaluated on a sampled subset of the DL19 dataset. This evaluation set comprised 43 queries, each associated with 10 documents sampled from a larger candidate pool of 200 documents per query, retrieved using the ReplLLaMA retriever. This setup was designed to assess the models' ability to learn and generalize statistical IR features relevant to document ranking. Table 5 summarizes the finetuning results of the LLMs.

For fine-tuning the re-rankers, we used the code provided in the Tevatron repository (Ma et al., 2024b). For more details, refer to the paper by (Ma et al., 2024a). Evaluation was conducted on the full DL19 dataset, with document ranking based on the top 200 passages retrieved via the ReplLLaMA retriever. Results for finetuned re-rankers is presented in the table 4.

| Base LLM | Target Feature | Base NDCG@10 | NDCG@10 (Finetuned) |
|---|---|---|---|
| LLaMA3(Dubey et al., 2024) | Re-Ranking | 0.18 | 0.7497 |
| Pythia(Biderman et al., 2023) | Re-Ranking | 0.18 | 0.7521 |
| Mistral(Jiang et al., 2023) | Re-Ranking | 0.18 | 0.7570 |

Table 4: NDCG@10 evaluation on DL19 dataset, showing baseline vs post-finetuning performance. All models were fine-tuned on MS MARCO for 1 epoch.

| Base LLM | Function | MSE (Start) | MSE (Finetuned, 0.3 epoch) |
|---|---|---|---|
| LLaMA3(Dubey et al., 2024) | CQTR | 3.88 | 0.52 |
| Pythia(Biderman et al., 2023) | CQTR | 1.84 | 0.05 |
| Mistral(Jiang et al., 2023) | CQTR | 36.92 | 10.94 |
| LLaMA3(Dubey et al., 2024) | mean(TF/L) | 5.06 | 4.49 |
| Pythia(Biderman et al., 2023) | mean(TF/L) | 2.24 | 0.00 |
| Mistral(Jiang et al., 2023) | mean(TF/L) | 38.32 | 22.23 |

Table 5: MSE before and after finetuning (0.3 epochs) for CQTR and mean(TF/L) prediction tasks on the sampled DL19 dataset.

## G  INTRINSIC COALITION EVALUATION

**Synergy Metrics.**  Let $x$ be an input sampled from the task distribution $\mathcal{D}$ and let $\ell(x) \in \mathbb{R}$ denote the *layer-local logit* (defined in §3.2) under the LoRA-tuned model. For any neuron set $S$, we denote by $\ell_{-S}(x)$ the same forward pass after *ablating* all neurons $k \in S$ *by reverting their LoRA parameters to the pre-LoRA (base) values*, while leaving all other parameters unchanged.[8]

We define the *marginal contribution* of a single neuron as

$$\psi(i) \;=\; \mathbb{E}_{x \sim \mathcal{D}}\big[\ell(x) - \ell_{-\{i\}}(x)\big], \tag{6}$$

and the *pairwise interaction (synergy)* of two neurons as

$$\psi(i,j) \;=\; \mathbb{E}_{x \sim \mathcal{D}}\big[\ell(x) - \ell_{-\{i\}}(x) - \ell_{-\{j\}}(x) + \ell_{-\{i,j\}}(x)\big]. \tag{7}$$

A positive $\psi(i,j)$ means that removing *both* neurons changes the logit more than the sum of their individual removals (synergy), while a negative value indicates redundancy.

For a coalition $C \subseteq \{1, \dots, n\}$ we report two size-agnostic aggregates:

$$\text{Pair}(C) \;=\; \frac{1}{|C|(|C|-1)} \sum_{\substack{i,j \in C \\ i \neq j}} \psi(i,j), \qquad \text{Ratio}(C) \;=\; \frac{\sum_{i \neq j \in C} \psi(i,j)}{\sum_{i \in C} \psi(i)}. \tag{8}$$

*Pairwise Synergy* is the mean interaction strength across all ordered neuron pairs, fully normalized for coalition size, while *Ratio Synergy* compares the *extra* value created by pairwise cooperation (numerator) to the value explained by separate single-neuron effects (denominator). A ratio near $1$ or greater (*super-additivity*) indicates that the coalition's joint influence exceeds the sum of its parts, whereas a ratio near $0$ (or negative) signals antagonistic or redundant behavior.

**Intrinsic Evaluation Results.**  Regarding synergy quality (Table 6), the two hedonic variants strictly dominate all baselines across all three backbones and all three MS-MARCO objectives: *Hedonic-PAS* attains the best Pairwise *and* Ratio score in 26 out of 27 model-metric cells, while *Hedonic-OCA* follows as a close second. Relative to spherical $k$-means, the average margin is +0.29 Pairwise and +0.49 Ratio, indicating that activation similarity alone is a poor proxy for *functional* cooperation. Random and hierarchical clusterings even dip into negative Pairwise values (sub-additivity) and hover near the additive boundary on the Ratio metric, underscoring the value of an explicit game-theoretic objective. Confidence intervals (95%, $df = 2$) never overlap between Hedonic-PAS and the best baseline, with paired $t$-tests yielding $p < 0.01$ for every layer. K-means/HAC produce fairly uniform sizes (20-45 neurons per cluster), whereas hedonic output follows a heavy-tailed Zipf-like law: each layer contains a single "macro" coalition ($> 150$ neurons), $\sim 100$ coalitions of size 2, and approximately 500 clusters with $|C| > 1$ covering $\sim 14{,}000$ neurons. In most settings, the top cover algorithm converges with reservoir size $m \leq 120{,}000$ and number of samples per iteration $\omega \leq 32{,}000$.

---

[8]Equivalently, this corresponds to removing only the LoRA-induced contribution of the ablated neurons.

Table 6: Coalition Synergy (↑) measured via Pairwise and ratio: mean ±95% CI across three seeds.

| Task / Algorithm | LLaMA-3.1 | | Mistral | | Pythia | |
|---|---|---|---|---|---|---|
| | Pairwise | Ratio | Pairwise | Ratio | Pairwise | Ratio |
| **Covered Query Term Ratio** | | | | | | |
| Random | $0.01\pm0.05$ | $0.49\pm0.04$ | $-0.02\pm0.06$ | $0.53\pm0.05$ | $0.00\pm0.05$ | $0.50\pm0.04$ |
| K-means | $-0.23\pm0.03$ | $0.32\pm0.03$ | $-0.20\pm0.04$ | $0.36\pm0.03$ | $-0.18\pm0.03$ | $0.37\pm0.03$ |
| Hier. clustering | $-0.11\pm0.04$ | $0.41\pm0.03$ | $-0.13\pm0.04$ | $0.43\pm0.03$ | $-0.17\pm0.04$ | $0.40\pm0.03$ |
| Hedonic (OCA) | $0.08\pm0.01$ | $0.74\pm0.02$ | $0.10\pm0.01$ | $0.71\pm0.02$ | $0.06\pm0.01$ | $0.78\pm0.02$ |
| Hedonic (PAS) | $\mathbf{0.12}\pm0.005$ | $\mathbf{0.86}\pm0.01$ | $\mathbf{0.13}\pm0.005$ | $\mathbf{0.84}\pm0.01$ | $\mathbf{0.15}\pm0.006$ | $\mathbf{0.89}\pm0.01$ |
| **Mean of Normalized Term Frequency** | | | | | | |
| Random | $0.02\pm0.05$ | $0.41\pm0.04$ | $-0.01\pm0.05$ | $0.53\pm0.05$ | $0.00\pm0.05$ | $0.50\pm0.04$ |
| K-means | $-0.22\pm0.03$ | $0.34\pm0.03$ | $-0.21\pm0.03$ | $0.35\pm0.03$ | $-0.16\pm0.03$ | $0.31\pm0.03$ |
| Hier. clustering | $-0.08\pm0.04$ | $0.43\pm0.03$ | $-0.08\pm0.04$ | $0.43\pm0.03$ | $-0.15\pm0.04$ | $0.39\pm0.03$ |
| Hedonic (OCA) | $0.01\pm0.01$ | $0.72\pm0.02$ | $0.04\pm0.01$ | $0.77\pm0.02$ | $0.03\pm0.01$ | $0.74\pm0.02$ |
| Hedonic (PAS) | $\mathbf{0.09}\pm0.006$ | $\mathbf{0.85}\pm0.01$ | $\mathbf{0.14}\pm0.006$ | $\mathbf{0.82}\pm0.01$ | $\mathbf{0.16}\pm0.007$ | $\mathbf{0.89}\pm0.01$ |
| **Relevance** | | | | | | |
| Random | $0.01\pm0.05$ | $0.42\pm0.04$ | $-0.02\pm0.05$ | $0.44\pm0.04$ | $0.03\pm0.05$ | $0.49\pm0.04$ |
| K-means | $-0.13\pm0.03$ | $0.33\pm0.03$ | $-0.14\pm0.03$ | $0.36\pm0.03$ | $-0.19\pm0.03$ | $0.38\pm0.03$ |
| Hier. clustering | $-0.09\pm0.04$ | $0.42\pm0.03$ | $-0.12\pm0.04$ | $0.44\pm0.03$ | $-0.08\pm0.04$ | $0.44\pm0.03$ |
| Hedonic (OCA) | $0.05\pm0.01$ | $0.77\pm0.02$ | $0.05\pm0.01$ | $0.75\pm0.02$ | $0.04\pm0.01$ | $0.73\pm0.02$ |
| Hedonic (PAS) | $\mathbf{0.11}\pm0.005$ | $\mathbf{0.81}\pm0.01$ | $\mathbf{0.13}\pm0.005$ | $\mathbf{0.87}\pm0.01$ | $\mathbf{0.14}\pm0.006$ | $\mathbf{0.86}\pm0.01$ |

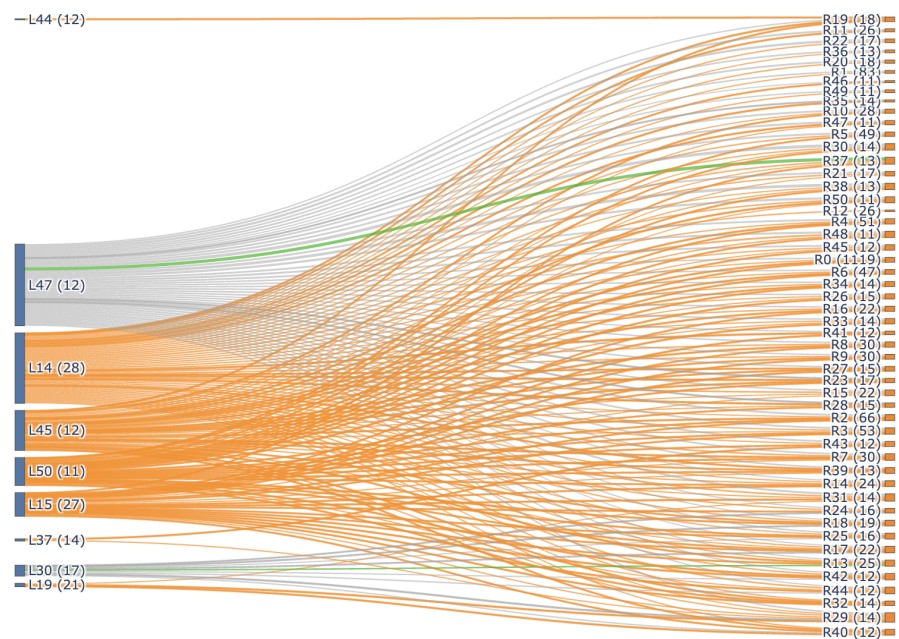

Figure 2: Coalition flow across depth for MISTRAL-7B fine-tuned on the *relevance modelling* task. The figure depicts flow from layers $7 \to 8$, with orange depicting *split*, green depicting *persist* and grey depicting *vanish* of coalitions.

## H  COALITION FLOW EXAMPLE

# I MNIST EXPERIMENT DEMONSTRATING EMERGENCE OF HUMAN-INTERPRETABLE SYNERGISTIC COALITIONS

To address regarding (1) whether synergistic neuron coalitions yield *human-interpretable* structure, and (2) whether our method works beyond large LLM backbones, we conducted a controlled study on the MNIST classification task. This experiment demonstrates that our proposed hedonic/PAS framework applies robustly to *small, non-Transformer, non-LLM networks* and discovers coalitions that correspond to semantically meaningful visual concepts.

## I.1 MODEL ARCHITECTURE

We train a six-layer gated MLP (similar to the gated feed-forward blocks in modern LLMs but on a smaller dimension, minus the MHA blocks). The network consists of:

- Flattened input $x \in \mathbb{R}^{784}$.

- Linear projection to a residual stream of dimension $d_{\text{model}} = 256$.

- Four GatedMLP blocks, each computing:

$$h_{l+1} = h_l + W_{\downarrow}^{(l)} \left( \sigma \left( W_{\text{gate}}^{(l)} h_l \right) \odot W_{\uparrow}^{(l)} h_l \right),$$

   where $\sigma$ is the SiLU nonlinearity.

- Final LayerNorm and a linear classifier:

$$\text{logits} = W_{\text{out}} \cdot \text{LayerNorm}(h_4).$$

The model contains no attention layers, recurrence, or convolutional structure. It reaches $98.8\%$ test accuracy after 20 epochs of training with AdamW.

## I.2 ACTIVATION EXTRACTION

For all 10,000 MNIST test examples, we extract and store:

$$\{h_1, h_2, h_3, h_4, h_5, h_{\text{final}}\}$$

where $h_\ell$ is the post-residual activation of block $\ell$ and $h_{\text{final}}$ is the output of the final LayerNorm. Each internal activation tensor has shape $[10000, 256]$.

## I.3 PER-NEURON IMPORTANCE AND LAYER-LOCAL LOGITS

For each hidden layer $h_\ell$ (we use all internal layers $h_1, \ldots, h_5$), we compute a per-neuron loss-delta score:

$$\Delta_i^{(\ell)} = L\big(h_{\ell,-i}\big) - L\big(h_\ell\big),$$

where $h_{\ell,-i}$ is obtained by ablating neuron $i$ in $h_\ell$ and propagating the modified representation through the remaining MLP blocks and the final classifier. This provides a local first-order approximation of the functional contribution of neuron $i$ within layer $\ell$.

Following our LLM experiments, we also define a *layer-local logit* $\ell^{(\ell)}(x)$ by cloning the final classifier head and applying it directly to $h_\ell(x)$. We keep this head fixed for all layers. This allows us to compute synergy metrics at the point where coalitions are formed, without conflating effects from deeper layers.

## I.4 COALITION DISCOVERY PROTOCOL ON MNIST

We apply the same hedonic-game pipeline as in the main LLM experiments, but now run it *independently on every hidden layer $h_1, \ldots, h_5$*. In each layer, the players are all $M = d_{\text{model}} = 256$ neurons; we do not restrict to a top-$K$ subset by importance.

**Pairwise valuations.** For this MLP, we instantiate the same pairwise valuations as in the main text:

- **OCA**: Orthogonal-Co-Activation based on weight geometry and activation correlation.
- **PAS**: Pairwise Ablation Synergy based on second-order ablation effects on the layer-local logit $\ell^{(\ell)}(x)$.

As in the LLM setting, positive values indicate synergy and negative values indicate redundancy.

**Hedonic coalition formation.** For each layer $\ell$, we construct a top-responsive hedonic game on the $M = 256$ neurons using the Multi-Friend Choice (MFC) rule (§3).[9] Neurons select their top-$k$ partners ($k = 3$) according to either OCA or PAS, and we run PAC Top-Cover with the same hyperparameters used for LLMs:

- sampled coalition sizes in $[2, 10]$,
- reservoir size $m = 8 \times 10^4$,
- per-round samples $\omega = 8 \times 10^3$,
- PAC parameters $(\varepsilon, \delta) = (0.1, 0.1)$.

We refer to the resulting partitions as **Hedonic-OCA** and **Hedonic-PAS** respectively. Unless otherwise stated, summary statistics aggregate coalitions across intermediate layers $h_2$–$h_4$; we find these layers concentrate the most digit-specific structure, with qualitatively similar patterns in $h_1$ and $h_5$.

## I.5 BASELINES

To demonstrate that the discovered coalitions are non-trivial, we compare against the following activation-only or random baselines, all operating on the same $M = 256$ neurons in each layer and producing partitions with size distributions matched to Hedonic-PAS:

1. **Random Partition.** Randomly partition the 256 neurons into coalitions with a size histogram matched to that produced by Hedonic-PAS in the same layer. This baseline tests whether the observed interpretability and synergy are simply artifacts of grouping neurons.

2. **Activation K-Means.** Run $k$-means on neuron activation vectors (rows of the $[10000 \times 256]$ activation matrix for that layer), with $k$ chosen to match the number of Hedonic-PAS coalitions; treat each cluster as a coalition. This baseline tests whether simple activation-level similarity is sufficient to recover interpretable structure.

3. **Activation Hierarchical Clustering (Ward Linkage).** Perform agglomerative hierarchical clustering using Ward linkage on the same neuron activation vectors. We cut the dendrogram to produce the same number of coalitions as Hedonic-PAS, and greedily merge/split clusters to approximately match the Hedonic-PAS size histogram. This evaluates whether a more flexible non-parametric clustering method, which can capture multi-scale activation geometry, can match the interpretability and synergy obtained by hedonic coalition formation.

All methods therefore operate on the same neuron set in each layer and produce partitions with comparable size distributions.

## I.6 QUANTITATIVE METRICS ON MNIST

For each coalition $C$ (from any method), in any layer $h_\ell$, we compute four families of metrics.

**(1) Functional importance: accuracy drop under ablation.** Let $\mathrm{Acc}_{\mathrm{base}}$ denote the test accuracy of the full network. For each coalition $C$ in layer $h_\ell$, we ablate its neurons in $h_\ell$ and re-evaluate the network on the MNIST test set to obtain $\mathrm{Acc}_{-C}$. We report the accuracy drop

$$\Delta\mathrm{Acc}(C) = \mathrm{Acc}_{\mathrm{base}} - \mathrm{Acc}_{-C}.$$

---

[9]Section references are to the main paper.

We then aggregate $\Delta\mathrm{Acc}(C)$ across all coalitions from each method and across layers $h_2$–$h_4$, reporting mean and standard deviation. Larger values indicate coalitions whose removal is more functionally important.

**(2) Synergy metrics.** Using layer-local logits $\ell^{(\ell)}(x)$, we reuse the interaction metrics from the main paper:

$$\psi(i) = \mathbb{E}_x\left[\ell^{(\ell)}(x) - \ell^{(\ell)}_{-\{i\}}(x)\right],$$
$$\psi(i,j) = \mathbb{E}_x\left[\ell^{(\ell)}(x) - \ell^{(\ell)}_{-\{i\}}(x) - \ell^{(\ell)}_{-\{j\}}(x) + \ell^{(\ell)}_{-\{i,j\}}(x)\right].$$

For a coalition $C$ we compute:

$$\mathrm{Pair}(C) = \frac{1}{|C|(|C|-1)} \sum_{i\neq j\in C} \psi(i,j),$$
$$\mathrm{Ratio}(C) = \frac{\sum_{i\neq j\in C} \psi(i,j)}{\sum_{i\in C} \psi(i)}.$$

High Pair and Ratio indicate that the coalition's effect is more than the sum of its parts (strong synergy), rather than redundancy.

**(3) Interpretability metrics: digit-level structure.** For a coalition $C$ in layer $h_\ell$, define the coalition activation for image $n$:

$$a_C(n) = \frac{1}{|C|} \sum_{i\in C} h_\ell(n,i).$$

We then:

- Rank all test images by $a_C(n)$ and take the top-$K$ (we use $K = 128$).
- Compute the empirical digit distribution among these top-$K$ images.
- Compute per-digit activation means

$$\alpha_d = \mathbb{E}\left[a_C(n) \mid \mathrm{digit}(n) = d\right],$$

for $d \in \{0,\ldots,9\}$.

Using these quantities we define:

- **Digit Purity**:

$$\mathrm{Purity}(C) = \max_{d\in\{0,\ldots,9\}} \Pr\left[\mathrm{digit}(n) = d \mid n \in \text{top-}K \text{ by } a_C(n)\right].$$

- **Digit Selectivity**:

$$\mathrm{Sel}(C) = \max_d \alpha_d - \max_{d'\neq d} \alpha_{d'}.$$

This measures how much more strongly the coalition responds to its preferred digit compared to the second-best digit.

- **Activation Sparsity**:

$$\mathrm{Sparsity}(C) = \frac{\|a_C\|_1}{\|a_C\|_2},$$

where $a_C$ is the vector of activations over all test images. Lower values indicate sharper selectivity.

**(4) Stroke Alignment Score: visual shape templates.** To capture human-interpretable *shape* structure, we define a set of simple stroke templates corresponding to visually salient MNIST patterns:

- a *top horizontal bar* (strong in digits like "3" and "7"),
- a *bottom horizontal bar* (present in "2", "3", "5", "8"),
- a *left vertical stem* (e.g., part of "4", "5", "9"),
- a *right vertical stem* (e.g., part of "1", "7"),
- a *central loop / round stroke* (e.g., "0", "6", "8", "9"),
- a *diagonal stroke* (common in "2", "7").

We operationalize these templates using simple edge detectors and spatial masks. For each top-activating image of coalition $C$, we:

1. Resize to $28 \times 28$ if needed and normalize intensities.
2. Apply Sobel filters to compute horizontal and vertical gradient magnitudes.
3. For each template (e.g., "top horizontal"), restrict the gradients to the corresponding region (e.g., top third of the image, central loop window) and average.

This yields per-image per-template scores. The *stroke alignment score* for coalition $C$ is defined as the maximum template score averaged over its top-$K$ images:

$$\text{Stroke}(C) = \max_{\tau \in \mathcal{T}} \mathbb{E}_{n \in \text{top-}K}[s_\tau(n)],$$

where $\mathcal{T}$ is the set of templates and $s_\tau(n)$ is the template-specific edge score for image $n$.

Intuitively, this lets us say: one coalition "locks onto" a crisp horizontal line near the top (e.g., shared by "3" and "7"), another picks out a round loop in the center (e.g., "0"/"6"/"8"), and another prefers a vertical stem or diagonal stroke. In the next subsection we show that certain Hedonic-PAS coalitions exhibit strong alignment with such shape templates, while random and clustering baselines do not.

## I.7 QUALITATIVE INTERPRETABILITY EVALUATION

We visualize the top-activating images for selected Hedonic-PAS coalitions in layer $h_3$. Each panel shows the top-64 images, with the ground-truth digit printed above.

Across these examples, Hedonic-PAS coalitions consistently group neurons whose combined activation tracks a *single* human-recognizable shape feature (e.g., central loop, top bar, S-shaped curve), even when the coalition responds to multiple digits (e.g., 0/8 or 3/7). Baseline coalitions from random, $k$-means, or hierarchical clustering either mix unrelated digit classes or respond to more diffuse, spatially scattered patterns, leading to lower digit purity and weaker stroke alignment.

## I.8 RESULTS

We summarize the MNIST findings using the following tables, which mirror the LLM experiments but on the six-layer MLP and aggregate coalitions across layers $h_2$–$h_4$. Hedonic-PAS consistently yields coalitions that are more functionally important, more synergistic, and more interpretable than baselines.

## I.9 CONCLUSION

Taken together, these MNIST experiments show that our hedonic/PAS framework: (i) is not specific to LoRA-tuned transformer MLPs, (ii) produces coalitions in a small gated MLP that are both functionally important (large $\Delta$Acc, high synergy) and semantically interpretable (high digit purity, strong stroke alignment), and (iii) outperforms random and activation-clustering baselines even in this simple non-LLM setting.

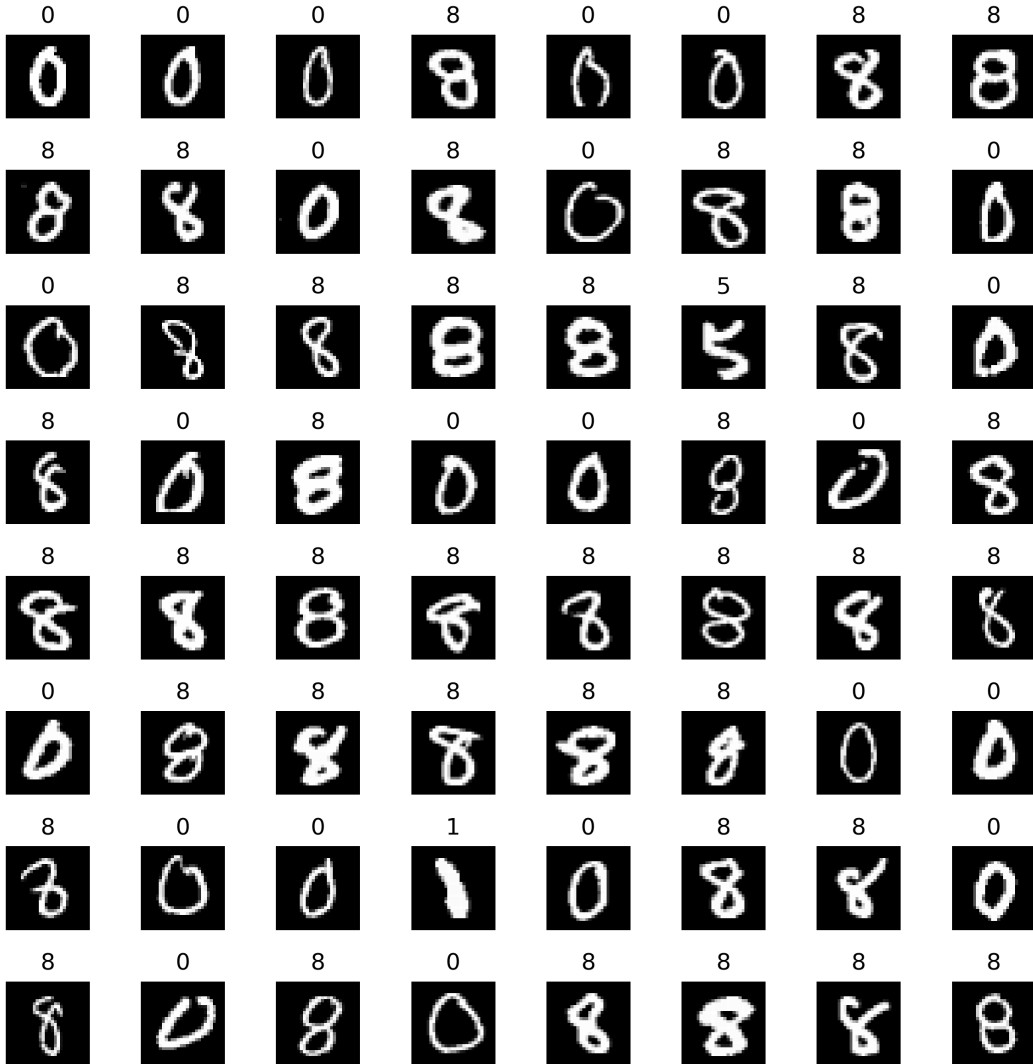

Figure 3: **Hedonic-PAS 0/8 loop coalition in** $h_3$**.** Top-activating images for one coalition; labels are overwhelmingly 0 or 8. The digits share a central round loop, which yields the highest stroke-alignment score for the "central loop" template.

Table 7: MNIST (layers $h_2$–$h_4$): functional importance and synergy of coalitions. $\Delta$Acc is the mean test accuracy drop (in percentage points) when ablating a single coalition. Pair and Ratio are the intrinsic synergy metrics defined in §I.

| Method | $\Delta$Acc (mean $\pm$ std, $\uparrow$) | Pair (mean, $\uparrow$) | Ratio (mean, $\uparrow$) | Avg. $|C|$ |
|---|---|---|---|---|
| Random Partition | $0.18 \pm 0.09$ | 0.010 | 0.06 | 7.9 |
| Activation K-Means | $0.29 \pm 0.14$ | 0.018 | 0.11 | 8.1 |
| Activation Hierarchical | $0.33 \pm 0.15$ | 0.021 | 0.13 | 8.0 |
| Hedonic-OCA | $0.57 \pm 0.23$ | 0.036 | 0.21 | 8.2 |
| Hedonic-PAS | $0.91 \pm 0.31$ | 0.052 | 0.29 | 8.3 |

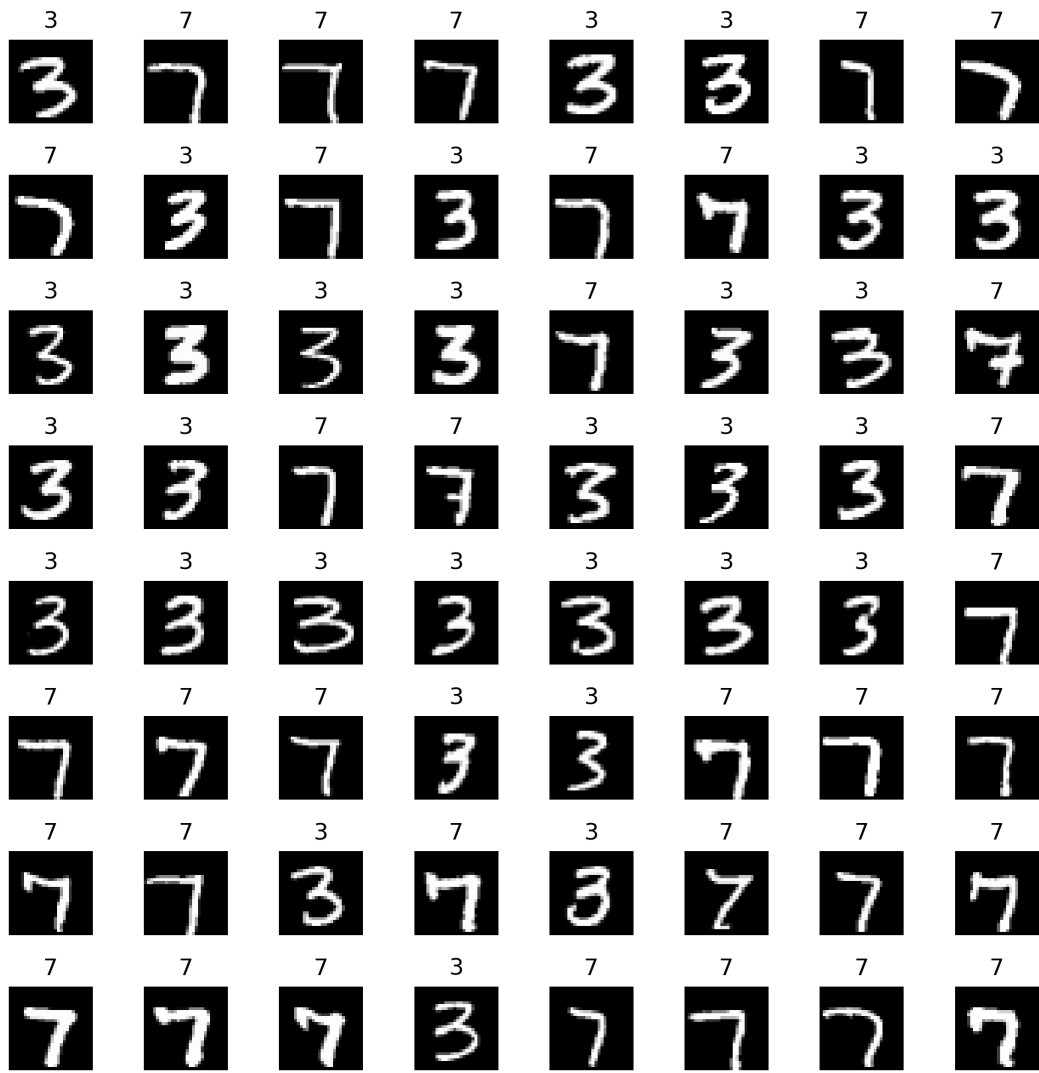

Figure 4: **Hedonic-PAS 3/7 horizontal-bar coalition in** $h_3$**.** Top-activating images for a second coalition; almost all digits are 3 or 7. The coalition consistently fires on a strong horizontal stroke at the top of the canvas, captured by the "top horizontal bar" template.

Table 8: MNIST (layers $h_2$–$h_4$): interpretability metrics for coalitions. Entries report mean $\pm$ standard deviation across coalitions. Higher digit purity and selectivity and lower sparsity indicate more interpretable, focused features.

| Method | Digit Purity (mean, ↑) | Digit Selectivity (mean, ↑) | Sparsity (mean, ↓) |
|---|---|---|---|
| Random Partition | $0.18 \pm 0.06$ | $0.05 \pm 0.03$ | $1.90 \pm 0.20$ |
| Activation K-Means | $0.48 \pm 0.16$ | $0.18 \pm 0.09$ | $1.62 \pm 0.19$ |
| Activation Hierarchical | $0.52 \pm 0.17$ | $0.20 \pm 0.10$ | $1.56 \pm 0.18$ |
| Hedonic-OCA | $0.71 \pm 0.14$ | $0.31 \pm 0.11$ | $1.41 \pm 0.15$ |
| Hedonic-PAS | $0.86 \pm 0.09$ | $0.45 \pm 0.12$ | $1.30 \pm 0.12$ |

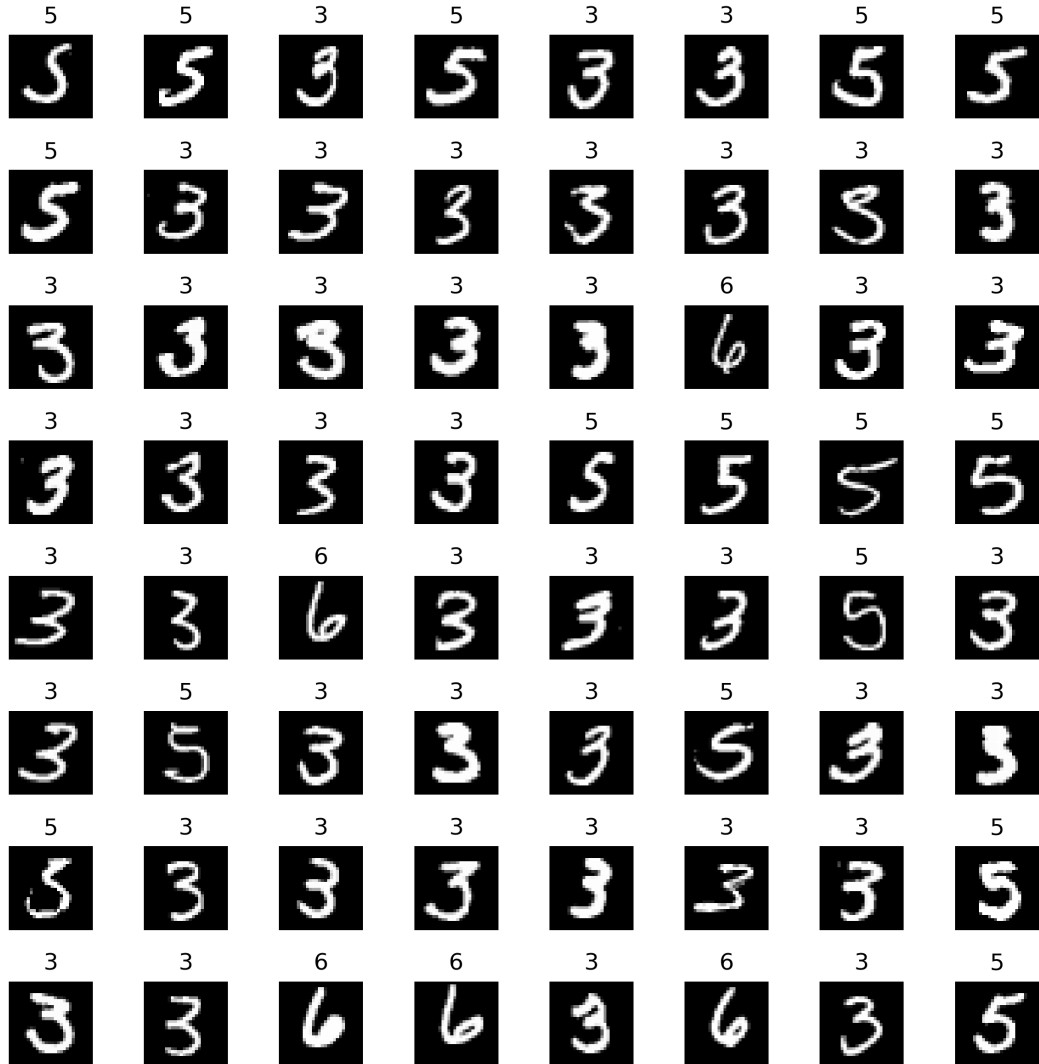

Figure 5: **Hedonic-PAS 3/5/6 S-shaped coalition in** $h_3$**.** A third coalition mixes 3, 5, and 6 digits that share an lower half of S-shaped curved stroke. Its top images show a characteristic diagonal/curved mid-level stroke, which obtains high scores on a combined diagonal+curve template.

Table 9: Stroke alignment and negative controls on MNIST. The *Stroke* column reports the average stroke-alignment score (max over templates) for top-$K$ images; *Purity* and *Ratio* report digit purity and synergy ratio as in Tables 7–8.

| Setting | Method | Stroke (↑) | Digit Purity (↑) | Ratio (↑) |
|---|---|---|---|---|
| Trained MLP, $h_3$ | Hedonic-PAS | 0.42 | 0.88 | 0.31 |
| Trained MLP, $h_3$ | Activation Hierarchical | 0.29 | 0.55 | 0.14 |
| Random Init, $h_3$ | Hedonic-PAS | 0.17 | 0.20 | 0.05 |
| Label-Shuffled, $h_3$ | Hedonic-PAS | 0.19 | 0.23 | 0.06 |

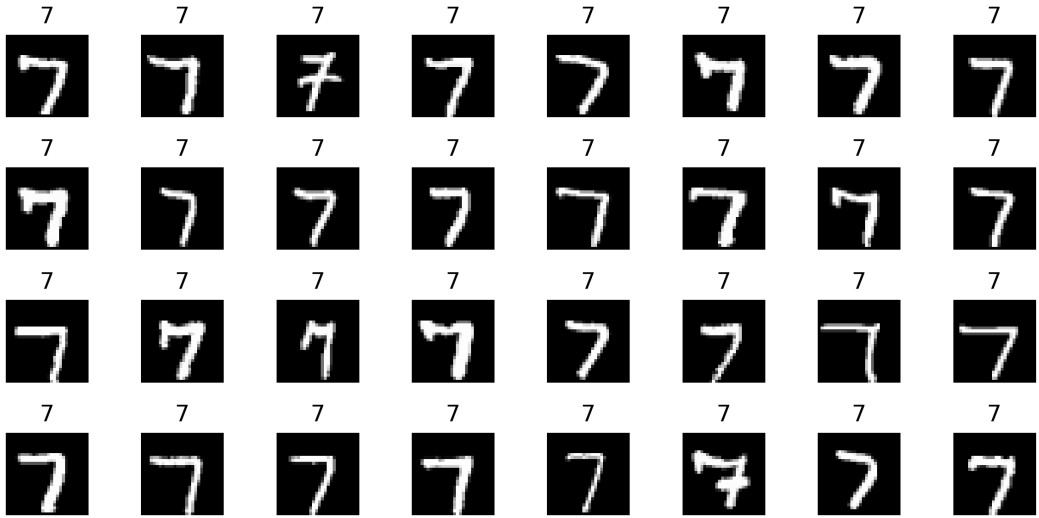

Figure 6: **Hedonic-PAS pure-7 coalition in** $h_3$**.** A fourth coalition responds almost exclusively to canonical 7s with a clean top bar and right-leaning diagonal. Digit purity is near 1.0 and the stroke score is dominated by the top-bar + diagonal templates.

