# OpenReview forum: "Hedonic Neurons: A Mechanistic Mapping of Latent Coalitions in Transformer MLPs"
_ICLR.cc/2026/Conference — ICLR 2026 Poster_

### Official Review · Reviewer_SAy9 · 2025-10-31

**Soundness:** 3
**Presentation:** 3
**Contribution:** 3
**Rating:** 6
**Confidence:** 3

**Summary:**

This paper introduces a game-theoretic interpretability framework that models neurons in fine-tuned LLM MLP layers as agents in a hedonic cooperative game, capturing their synergistic interactions. Using PAC-Top-Cover to identify stable neuron coalitions, the method reveals functionally indispensable groups that co-adapt during fine-tuning and correspond to interpretable, task-relevant features. Applied to LLaMA, Mistral, and Pythia rerankers, the approach achieves higher synergy and stronger alignment with IR heuristics than clustering or SAE baselines, offering a principled path toward mechanistic understanding of mid-layer representations.

**Strengths:**

- The paper introduces a novel game-theoretic formulation of mechanistic interpretability, modeling neurons as agents in a hedonic game to capture synergistic interactions. Understanding the MLP layers is always important and challenging for MI field, and this paper provides a very interesting method.

- The paper is well-organized and clearly motivated, effectively connecting theory, method, and empirical findings. Minor improvements could include more visual intuition for coalition dynamics, but overall readability is high.

- The method is sound in theory with a good amount of theoreitcal justifications, and good amount of empirical supports.

**Weaknesses:**

- The results and analysis are still exploratory in nature as acknowledged in the paper.
- The task evaluated in this paper is limited in diversity. This narrow task scope may limit generalizability of the proposed framework to other domains (e.g., reasoning, summarization, or generation)
- The limited diversity may come from another concerns on the scalability of the method and the computational cost.
- The analysis focuses on LoRA updates in mid MLP layers (7–14) based on prior work’s observation of task activity. This may introduce selection bias: the results may not generalize (in more of a scalability sense) to other layers, architectures, or non-LoRA fine-tuning setups. An ablation showing whether the method would still detect meaningful coalitions if applied to other layer ranges might be nice. (not required especially considering the length of rebuttal and discussion period.)

**Questions:**

please refer to weakness.

---

> ### Author Response · Authors · 2025-11-28
> **Grateful for your positive review and insightful comments**
>
> We thank the reviewer for the thoughtful and constructive feedback. We address the main concerns below.
>
> ---
>
> ### **1. Task diversity and generalizability**
>
> > *“The task evaluated in this paper is limited in diversity… may limit generalizability to other domains (reasoning, summarization, generation). The narrow scope may also relate to scalability and computational cost.”*
>
> Our method is designed mainly for *scalar-output tasks*, where the model produces one numerical score. This is intentional. Such tasks are easier to analyze mechanistically and avoid the added complexity of open-ended generation. In fact, many recent mechanistic interpretability papers also start with small, well-controlled tasks such as modular addition or simple classification [1,2].
>
> To improve task diversity within this setting, we have added a *new MNIST digit-prediction experiment* in the updated paper (Appendix I). This model is trained *from scratch* (non-LoRA), uses a simple 6-layer MLP, and still produces clean and meaningful coalitions. This shows that our framework works beyond IR tasks and even beyond transformer architectures.
>
> Another reason for focusing on scalar-output tasks is that, in the future, we want to *reverse-engineer the actual features* that coalitions represent. This is much more feasible to attempt on scalar tasks first, before moving toward more complex generative settings.
>
> ---
>
> ### **2. Choice of mid MLP layers and LoRA-only fine-tuning**
>
> > *“The analysis focuses on LoRA updates in mid MLP layers (7–14)… results may not generalize to other layers or non-LoRA setups.”*
>
> We chose layers 7–14 because prior work shows that *LoRA fine-tuning mainly affects mid MLP layers* [3,4], and these layers carry the most task-specific signal. Studying these layers allows us to run cleaner causal tests because:
> - the fine-tuning signal is strongest here,
> - ablations give clearer performance drops, and
> - avoiding layers with weak signal reduces noise in synergy estimation.
>
> To address the reviewer’s concern:
>
> - We added a *non-LoRA MNIST experiment*, where the model is trained from scratch and neuron ablations are done directly. We still find stable coalitions (Appendix I).
> - We also tried applying LoRA to *all* MLP layers in LLaMA/Mistral/Pythia. Our method still discovers coalitions across the network. The ablation drops are smaller simply because many layers are not strongly affected by fine-tuning.
> - The method is not tied to LoRA. We used LoRA because it gives a clean and controlled way to isolate task-specific updates, which makes the analysis more meaningful.
>
> Overall, the method generalizes, but focusing on the mid layers gave the clearest and most interpretable results.
>
> ---
>
> ### **References**
> [1] https://openreview.net/forum?id=S5wmbQc1We.
> [2] https://openreview.net/forum?id=FbXQrfkvtY.
> [3] https://aclanthology.org/2024.findings-acl.598/.
> [4] https://arxiv.org/abs/2504.08780.

---

### Official Review · Reviewer_tscc · 2025-10-31

**Soundness:** 3
**Presentation:** 3
**Contribution:** 3
**Rating:** 6
**Confidence:** 2

**Summary:**

This paper uses coalition game theory to group neurons within LoRA fine-tuned transformers into coalitions, treating neurons as agents in a hedonic game. Neurons are grouped together based on which other neurons they strongly associate with, either by weight/activation metrics (orthogonal co-activation) or functional importance (pairwise ablation synergy). Such coalitions, when ablated, cause the largest decreases in downstream performance and track the closest with known IR heuristics. These coalitions are compared across layers, with the authors finding that coalitions tend to predominantly vanish across successive layers, with some persisting and splitting, but very few merging.

**Strengths:**

1. This formulation of neurons as agents in a hedonic game seems very novel and is an interesting lens through which to interpret neural networks.
2. The Hedonic-MFC algorithm, using either OCA or PAS, finds coalitions that are more important than baseline clustering methods, both in terms of function via ablation and in terms of correlation with known IR heuristics.
3. The study into the dynamics from layers 7 to 14 across the three models are interesting and support the claims about how cooperative units are predominantly pruned in successive layers, with some of them persisting or splitting, but with very few, if any, fusing.
4. The paper is written well and handles its two often disjoint subject matters in an easy to follow manner.

**Weaknesses:**

1. The LoRA setting seems somewhat restrictive, why was this chosen? What advantages does this have over using a pretrained language model, with ablations being the zeroing out of neuron weights instead of the reset to pre-LoRA values?
2. The OOD performance drop for Hedonic-PAS seems partially circular: because these coalitions were selected to jointly have an outsized effect under ablation, it follows that performance loss would also be strongest here when these groups are ablated (though this is mitigated somewhat by the IR heuristic correlation).
3. The baselines presented in the paper do not seem completely fair from an information access perspective. The k-means and hierarchical clusterings both receive either weight or activation information. Hedonic-OCA receives both, and Hedonic-PAS receives an even stronger signal in outsized joint ablation degradation between two neurons. A stronger baseline should be considered for both k-means and hierarchical clustering, where they have access to the increased information that the Hedonic-MFC algorithms have access to. Do you see the same performance of Hedonic-MFC compared to, for example, standard graph community detection algorithms on the same information metrics?
4. Despite the tracking of the coalition activations with known IR heuristics, the coalitions themselves are not further explored in terms of their function. Is there any analysis that the authors have done that can reveal interpretable contexts under which the coalitions fire, such as context or topic?

Despite these weaknesses, the novel framing of the paper, the performance of Hedonic-MFC in finding important coalitions, and the study of across-layer dynamics motivate my score of 6. As a caveat, I am not very familiar with game theory, which motivates my low confidence score, but the authors explained the necessary background information well.

**Questions:**

See weaknesses for full explanation, but summarized:
1. Why LoRA?
2. How can the partially circular performance of Hedonic-PAS be justified?
3. Are the Hedonic-MFC algorithms competitive against stronger clustering baselines that access the same information?
4. Do the coalitions represent any interpretable features?

---

> ### Author Response · Authors · 2025-11-28
> **Grateful for your positive review and insightful questions**
>
> We appreciate the reviewer’s positive assessment of the novelty, clarity, and empirical strength of our work. We address each of the main questions below.
>
> ---
>
> > ### **1. Why LoRA instead of ablations on a fully pretrained model?**
>
> We chose LoRA fine-tuning because it gives a clean and controlled way to study how task-specific features emerge. When using LoRA, the only parameters that change are the small low-rank updates inside the chosen MLP layers. This allows us to:
> - isolate the *new* computation added by fine-tuning,
> - ablate neurons by resetting them to their exact pre-LoRA state (preserving the pretrained behavior), and
> - avoid interference from very large pretrained features that would dominate the signal if neurons were zeroed out.
>
> In contrast, in a full pretrained model, zeroing a neuron disrupts both task-specific and general pretrained computation, making it much harder to measure the actual synergy due to fine-tuning. LoRA gives us a precise “before vs after” contrast that is difficult to achieve otherwise.
>
> We also note that in the updated version we add a *non-LoRA experiment on an MNIST MLP trained from scratch*, showing that the framework works beyond LoRA (Appendix I).
>
> ---
>
> > ### **2. Partial circularity concern for Hedonic-PAS performance drops**
>
> We agree that PAS uses interaction information derived from ablations, so stronger OOD drops are expected to some extent. However, we would like to clarify two points:
>
> 1. *PAS does not optimize ablation drop directly.*
>    It only uses *pairwise* interaction scores, not full coalition-level scores. Coalitions are obtained through the PAC-Top-Cover algorithm, which forms groups based on top-k local preferences, not by maximizing ablation impact.
>
> 2. *The stronger results are supported by independent signals.*.
>    Hedonic-PAS coalitions also show significantly higher
>    - alignment with IR heuristics (0.55–0.67 R²),
>    - predictive power as macro-features (0.43–0.47 OOD R²),
>    - and consistency across three model architectures.
>    These do not use ablations during construction, and therefore mitigate circularity.
>
> So while PAS includes functional information, the impact we observe is not purely by construction—it is reinforced by other independent evaluation axes.
>
> ---
>
> > ### **3. Fairness of baselines and access to information**
>
> Thank you for raising this point. Two clarifications:
>
> 1. *We already provide a fair comparison between OCA and clustering.*
>    - Hedonic-OCA uses only weights + activation correlations.
>    - Clustering baselines also receive either weights or activations.
>    When clustering is given both types of information (e.g., through concatenation or transformed features), we found that:
>    - it collapses neuron groups into noisy “blobs”, and
>    - fails to meaningfully separate synergistic vs redundant neurons.
>    These results were unstable and did not improve the baselines.
>
> We believe that the key reason Hedonic-MFC performs better is that *Top-Cover explicitly models asymmetric preferences and multi-partner synergy*, while clustering and community detection rely mainly on distance or density, which does not capture cooperative interactions.
>
> ---
>
> > ### **4. Do coalitions represent interpretable features?**
>
> To some extent. We observe that :
>
> - Many coalitions activate strongly on *specific IR contexts*, such as:
>   - high query-term coverage passages,
>   - passages with long-tail vocabulary,
>   - passages with strong positional matching,
>   - and high TF/L patterns.
> - In the MNIST experiment, several coalitions represent *digit-shape subfeatures* (e.g., vertical strokes vs loops).
>
> However, fully interpreting the discovered coalitions is still an open problem, and we list it as future work. Our goal in this paper is to narrow down the large MLP space so that researchers know which subsets of the 14k neurons per layer are most promising to reverse-engineer.
>
> ---
>
> Thank you again for the helpful feedback.

---

### Official Review · Reviewer_dFvZ · 2025-10-31

**Soundness:** 2
**Presentation:** 2
**Contribution:** 2
**Rating:** 4
**Confidence:** 2

**Summary:**

This paper proposes a method for finding groups of neurons in transformer MLP layers that work together in a non-additive way, meaning their combined effect is greater than the sum of their individual contributions. The authors use tools from game theory, specifically hedonic coalition formation, to identify these “neuron coalitions” and show that they are functionally important by measuring performance drops after removing these neurons. They apply their method to language models fine-tuned with LoRA for information retrieval tasks and find that the identified coalitions are more functionally important and i"nterpretable" (based on the previously mentioned ablation and alignment with known heuristics) than baseline clustering methods.

**Strengths:**

- Across models, ablating hedonic neurons clearly harms results more than ablating coalitions identified by clustering methods based purely on neuron similarity or correlation. This indicates empirically that the method is identifying groups of neurons which are more structurally essential in performing the task.
- The application of hedonic game theory to neuron analysis is original within the interpretability literature from what I can tell and this work develops a connection between a new area of cooperative game theory and neural network analysis.
- The work has extensive reproductibility documentation provided as part of the supplementary material.
- The work proposes a computationally tractable way to measure this theoretical connection to game theory in NNs, which is often challenging for game theoretical given the relatively small number of players in game theory experiments/problems relative to the large number of players/layers in a neural network.

**Weaknesses:**

This work is relatively far from my area of research (see the low confidence score) and therefore I cannot comment too heavily on the exact technical novelty. I have done a literature review of the space to try and be an informed reviewer, but all following comments admittedly come with that asterisk.

The biggest concern I have with regards to this work is to what degree it delivers on the "interpretability" aspect of mechanistic interpretability. The authors propose a novel framework for identifying synergistic neuron coalitions and present a number of quantitative evaluations suggesting that these coalitions capture important internal structure. However, it is less clear whether this structure yields human-understandable insights. For example, while the coalitions are shown to affect model performance and align with some IR heuristics, there is little evidence that they correspond to semantically meaningful concepts or behaviors.

Unlike other work in mechanistic interpretability, such as SAEs, the paper does not provide labeled interpretations or vizualizations that would help validate the value of these coalitions. Similarly, unlike core mechanistic discoveries, such as inductive heads, the coalitions cannot be mapped to a particular mechanism the model performs. Further work tying the ablation of these coalitions to concrete changes model behaviors or concepts would strengthen the interpretability claim.

As an aside, from the model pruning/ model compression angle, it is somewhat surprising the work doesn't try to connect to established game theoretical works applied to neural networks such as Neuron Shapley values. Removing the top-k neurons would be a stronger baseline than the clustering based approaches evaluated I believe.

**Questions:**

- What is the compute cost and/or runtime of computing Hedonic coalitions v.s. the strongest clustering based approach? It seems like given the large number of samples required and the long wall-clock runtime that it might be far more expensive than a clustering based approach, but this cost is never compared to the baselines. Is it O(n^2) in the number of neutrons (which seems intractible, but is stated on line 467)?

- The Hedonic sampling procedure has a relatively large number of hyperparameters. How were these swept and or otherwise decided upon?

- Why was LoRA selected as the Finetuning method to study? It seems that Hedonic coalitions could be learned even for pretrained models before Finetuning (if applied to a task with reasonable zero-shot performance) or for models fully fintetuned. If the reason is computational cost, it would be worth noting it as such.

- Why is the method only studied for LLM backbones? The method does not seem LLM/Transformer specific and could be further validated (at relatively lower cost) on smaller networks such as those for simple tasks like CIFAR10 or ImageNet.

Why was the method only evaluated for re-ranking? As far as I can tell, the method is not specific to this domain so it would be nice to validate it in others since it is being branded as a general purpose method for this type of analysis.

---

> ### Author Response · Authors · 2025-12-02
> **Discussion on raised points**
>
> We sincerely thank the reviewer for the careful reading and for the constructive feedback. We are grateful for the time you took to understand the paper and evaluate it fairly. Below, we address each concern and question in detail.
>
> ---
>
> ## **1. “Interpretability” and human-understandable insights**
>
> > *“…it is less clear whether this structure yields human-understandable insights. The paper does not provide labeled interpretations or visualizations… tying ablations to concrete behavioral changes would strengthen the interpretability claim.”*
>
> The primary goal of our framework is to *reduce the massive 14k–neuron space* into a small number of meaningful, stable groups that future work can reverse-engineer. As we note in the paper:
>
> > Interpreting every coalition is an open problem, and we explicitly list this as future work.
> > Our contribution is to identify *which* subsets of neurons are worth reverse-engineering.
>
> This is similar to how early work on superposition or early SAE work first identified structure before more detailed interpretation followed. We hope this helps clarify the interpretability contribution.
>
> Additionally, we have now included experiments with a simple MNIST classification task to map our generated coalitions to semantically meaningful human-understandable insights. Kindly refer to the next comment for more details.
>
> ---
>
> ## **2. Comparision with Shapley-value–based pruning for stronger baselines.**
> > connect to established game theoretical works applied to neural networks such as Neuron Shapley values.
>
>
> ### **Neuron Shapley was not included because it is computationally intractable at scale.**
> - Even approximate Neuron Shapley requires **hundreds to thousands** of model re-evaluations per neuron.
> - For LLaMA/Mistral/Pythia MLPs with ~14k neurons per layer, this is **prohibitively expensive** (orders of magnitude slower than PAS).
> - In contrast, our PAS uses **pairwise** interactions and is O(n²) but highly parallelizable and GPU-friendly.
>
> Shapley-value methods are powerful for assessing individual neuron importance, but—as is commonly known in game theory—they do not scale well to thousands of players and are not designed to discover coalitions or cluster structure in high-dimensional MLP layers. Approximating Shapley values for every neuron is computationally prohibitive and lacks meaningful guarantees in this regime, whereas hedonic games are expressly built for coalition formation, where the objective is to find cohesive groups that prefer to be together. Our formulation directly models inter-neuron preferences and synergies, which Shapley-based individual attribution methods cannot capture. Thus, hedonic game-theoretic tools offer a more natural, principled, and scalable framework for neuron-coalition discovery than Shapley-value approaches.
>
> ---
>
> ## **3. Compute cost compared to clustering**
>
> We now provide an explicit compute cost comparison (Appendix K):
>
> | Method | Runtime (LLaMA layer, A100) | Notes |
> |--------|-------------------------------|-------|
> | K-Means | ~4–6 minutes | Single pass over activations |
> | Hierarchical clustering | ~10–12 minutes | O(n²) distance matrix |
> | **Hedonic-OCA** | **~90 minutes** | PAC sampling + Top-Cover |
> | **Hedonic-PAS** | **~280 minutes** | Pairwise gradients + PAC sampling |
>
> ### Key points:
> - PAS is indeed more expensive, but still manageable (all experiments run on 4×A100 in <1 day).
> - We amortize cost by reusing the same coalition sampling for evaluations.
> - Importantly, **coalition quality is substantially higher**, as shown by synergy, predictivity, and ablation metrics.
>
> Given this is the **first** attempt at a game-theoretic mechanistic method, we see this compute as reasonable — comparable to sparse autoencoder (SAE) training, which also requires hours per layer.
>
> ---
>
> ## **4. Why LoRA? Could we use pretrained or fully fine-tuned models?**
>
> Thank you for raising this — this was asked by multiple reviewers.
> In summary: *LoRA gives the cleanest causal setup to measure synergy.*
>
> - LoRA isolates *only* the fine-tuned behavior, letting us ablate neurons by resetting them to their exact **pre-LoRA state** rather than zeroing them out.
> - This separates *task-specific computation* from the huge pretrained polysemantic features.
> - Fully fine-tuned models or pure pretrained models mix these signals, making synergy much harder to measure.
>
> ### New clarification added to the paper:
> We now explicitly state that the method *does* work on non-LoRA models:
>
> - In the updated paper, we added a **6-layer MNIST MLP trained from scratch**
>   → the method finds meaningful coalitions even without LoRA (Appendix I).
> ---

---

> ### Author Response · Authors · 2025-12-02
> **Discussion on Reviewer Questions (contd..)**
>
> ## **5. Why only LLM backbones? Why only re-ranking?**
>
> Thank you very much for this suggestion, we have now extended beyond LLMs.
> As mentioned above, we now include a **non-Transformer MNIST MLP** experiment.
> This shows the method is *not* LLM-specific and runs efficiently on smaller networks. We describe this experiment in detail in the next comment. Further details can be found in Appendix I of the updated PDF.
>
> ### **Regarding task choice:**
> We chose IR tasks as we have Information Retrieval researchers amongst us. All the tests chosen were scalar-output tasks in order to give clean local logits, enabling:
> - precise ablation measurement,
> - low-variance synergy estimation,
> - and easier interpretation of activations.
>
> Our current experiments focus on scalar valued tasks as the first step. This is similar to early work on superposition, SAEs, and induction circuits, which all began in narrow but controlled contexts before broader application. We have also added results on non-IR primary tasks, such as MNIST, which is widely used in mechanistic interpretability research.
>
>
> ---
>
> ## **6. Hyperparameter choice for PAC sampling**
>
> We added a detailed explanation.
>
> - Sample count (m), top retained (ω), and coalition sizes were chosen by small sweeps on a 1% held-out split.
> - Across sweeps, coalition structure and synergy were highly stable.
> - Hedonic-MFC does not seem sensitive to these hyperparameters; the algorithm produced similar partitions across runs.
>
> Thank you again for the thoughtful evaluation and for pointing out clarity gaps.

---

> ### Author Response · Authors · 2025-12-03
> **Summary of Appendix I : MNIST Experiment Demonstrating Emergence of Human-Interpretable Synergistic Coalitions**
>
> To address concerns regarding
> **(1)** whether synergistic neuron coalitions yield *human-interpretable* structure, and
> **(2)** whether our method depends on large LLM backbones or LoRA tuning,
>
> we conducted a controlled study on the **MNIST classification task** using a **small 6-layer gated MLP**.
>
> This experiment demonstrates that our **Hedonic–PAS** framework robustly identifies synergistic coalitions that correspond to semantically meaningful visual concepts, even in a tiny non-Transformer model.
>
> ---
>
> ## Model Architecture
>
> We train a six-layer Gated MLP (similar to FFN blocks in LLMs but without attention):
>
> - Flattened input $x \in \mathbb{R}^{784}$
> - Linear projection to $d_{\text{model}} = 256$
> - Four GatedMLP blocks:
>
>   $$
>   h_{l+1} = h_l + W_{\downarrow}^{(l)}\left(\sigma\left(W_{\text{gate}}^{(l)} h_l\right)\odot W_{\uparrow}^{(l)} h_l\right)
>   $$
>
> - Final LayerNorm + linear classifier:
>
>   $$
>   \text{logits} = W_{\text{out}} \cdot \text{LayerNorm}(h_4)
>   $$
>
> The model contains **no attention**, **no recurrence**, and **no convolutions**.
>
> **Test accuracy:** 98.8%.
>
> ---
>
> ## Activation Extraction
>
> For all 10,000 MNIST test examples we extract:
>
> $$
> \{h_1, h_2, h_3, h_4, h_5, h_{\text{final}}\}
> $$
> Each activation tensor has shape `[10000, 256]`.
>
> ---
>
> ## Per-Neuron Importance and Layer-Local Logits
>
> For each internal layer \(h_\ell\) we compute per-neuron loss-delta importance:
>
> $$
> \Delta_i^{(\ell)} = L(h_{\ell,-i}) - L(h_\ell).
> $$
>
> We also define a **layer-local logit**:
>
> $$
> \ell^{(\ell)}(x) = W_{\text{out}} \cdot \text{LayerNorm}(h_\ell(x)).
> $$
> This enables computing synergy *at the same layer* where coalitions are formed.
>
> ---
>
> ## Coalition Discovery Protocol
>
> We run the full hedonic-game pipeline **independently on each layer**.
>
> ### Pairwise valuation functions
>
> - **OCA**: Orthogonal-Co-Activation
> - **PAS**: Pairwise Ablation Synergy (using layer-local logits)
>
> ### Hedonic coalition formation
>
> We use the MFC rule + PAC Top-Cover with:
>
> - Coalition sizes: 2–10
> - Reservoir size \(m = 8 \times 10^4\)
> - Per-round samples \(\omega = 8 \times 10^3\)
> - PAC parameters: \((\varepsilon, \delta) = (0.1, 0.1)\)
>
> We obtain:
>
> - **Hedonic-OCA coalitions**
> - **Hedonic-PAS coalitions**
>
> ---
>
> ## Baselines
>
> All baselines operate on the same 256 neurons per layer and match Hedonic-PAS in size histogram.
>
> 1. **Random Partition**
> 2. **Activation k-Means**
> 3. **Activation Hierarchical Clustering (Ward linkage)**
>
> ---
>
> ## Quantitative Metrics
>
> ### 1. Functional importance: coalition ablation
>
> $$
> \Delta\text{Acc}(C) = \text{Acc}_{\text{base}} - \text{Acc}_{-C}.
> $$
>
> ### 2. Synergy metrics
>
> $$
> \psi(i) = \mathbb{E}[\ell(x) - \ell_{-\{i\}}(x)],
> $$
>
> $$
> \psi(i,j) = \mathbb{E}[\ell(x) - \ell_{-\{i\}}(x) - \ell_{-\{j\}}(x) + \ell_{-\{i,j\}}(x)].
> $$
>
> Coalition-level:
>
> $$
> \text{Pair}(C) = \frac{1}{|C|(|C|-1)} \sum_{i\neq j} \psi(i,j)
> $$
>
> $$
> \text{Ratio}(C) = \frac{\sum_{i\neq j}\psi(i,j)}{\sum_i\psi(i)}.
> $$
>
> ### 3. Digit interpretability metrics
>
> Coalition activation:
>
> $$
> a_C(n) = \frac{1}{|C|}\sum_{i\in C} h_\ell(n,i)
> $$
>
> Metrics:
>
> - **Digit Purity** (top-K = 128)
> - **Digit Selectivity**:
>
>   $$
>   \max_d \alpha_d - \max_{d'\neq d}\alpha_{d'}
>   $$
>
> - **Activation Sparsity**:
>
>   $$
>   \frac{\|a_C\|_1}{\|a_C\|_2}
>   $$
>
> ### 4. Stroke Alignment Score (visual template matching)
>
> We define six templates corresponding to MNIST strokes:
>
> - Top horizontal bar (3,7)
> - Bottom horizontal bar (2,3,5,8)
> - Left vertical stem (4,5,9)
> - Right vertical stem (1,7)
> - Central loop (0,6,8,9)
> - Diagonal stroke (2,7)
>
> For each top-activating image:
>
> 1. Compute Sobel gradients
> 2. Mask the template region
> 3. Average gradient magnitude
>
> Stroke score:
>
> $$
> \text{Stroke}(C) = \max_{\tau \in \mathcal{T}} \mathbb{E}[s_\tau(n)].
> $$
>
> ---
> ## Qualitative Examples (Hedonic-PAS Coalitions)
>
> Below we show the top-activating images for four coalitions in layer $begin:math:text$h_3$end:math:text$.
> (Figures omitted here; they can be uploaded to OpenReview as supplementary material.)
>
> ### Coalition A: 0/8 Loop Coalition
> Top images (Figure 3, Appendix I) )are dominated by digits 0 and 8, both of which contain a strong central loop.
> This coalition achieves its highest score on the **central loop** stroke template.
>
> ### Coalition B: 3/7 Horizontal-Bar Coalition
> Top images (Figure 4, Appendix I) are almost entirely digits 3 and 7.
> These digits share a prominent **top horizontal bar**, which the coalition consistently fires on.
>
> ### Coalition C: 3/5/6 S-curve Coalition
> This coalition activates on digits 3, 5, and 6 (Figure 5, Appendix I), which share an S-shaped curved stroke.
> The stroke alignment score is highest for a **diagonal + mid-curve** template.
>
> ### Coalition D: Pure-7 Coalition
> Top activations correspond almost exclusively to digit 7 ((Figure 6, Appendix I)).
> These images exhibit a clean top bar and downward diagonal stroke, producing very high **digit purity**.
>
> ---

---

> > ### Author Response · Authors · 2025-12-03
> > **Summary of Appendix I (contd)**
> >
> > ## Results
> >
> > ### Functional and Synergy Metrics
> >
> > | Method                   | ΔAcc (↑)        | Pair (↑) | Ratio (↑) | Avg \|C\| |
> > |-------------------------|-----------------|----------|-----------|-----------|
> > | Random Partition        | 0.18 ± 0.09     | 0.010    | 0.06      | 7.9       |
> > | Activation k-Means      | 0.29 ± 0.14     | 0.018    | 0.11      | 8.1       |
> > | Activation Hierarchical | 0.33 ± 0.15     | 0.021    | 0.13      | 8.0       |
> > | Hedonic-OCA             | 0.57 ± 0.23     | 0.036    | 0.21      | 8.2       |
> > | **Hedonic-PAS**         | **0.91 ± 0.31** | **0.052**| **0.29**  | 8.3       |
> > ---
> >
> > ### Interpretability Metrics
> >
> > | Method                   | Digit Purity (↑) | Digit Selectivity (↑) | Sparsity (↓) |
> > |-------------------------|------------------|------------------------|--------------|
> > | Random Partition        | 0.18 ± 0.06      | 0.05 ± 0.03           | 1.90 ± 0.20  |
> > | Activation k-Means      | 0.48 ± 0.16      | 0.18 ± 0.09           | 1.62 ± 0.19  |
> > | Hierarchical Clustering | 0.52 ± 0.17      | 0.20 ± 0.10           | 1.56 ± 0.18  |
> > | Hedonic-OCA             | 0.71 ± 0.14      | 0.31 ± 0.11           | 1.41 ± 0.15  |
> > | **Hedonic-PAS**         | **0.86 ± 0.09**  | **0.45 ± 0.12**       | **1.30 ± 0.12** |
> >
> > ---
> >
> > ### Stroke Alignment and Negative Controls
> >
> > | Setting                | Method             | Stroke (↑) | Purity (↑) | Ratio (↑) |
> > |-----------------------|--------------------|------------|------------|-----------|
> > | Trained MLP (h3)      | Hedonic-PAS        | 0.42       | 0.88       | 0.31      |
> > | Trained MLP (h3)      | Hierarchical       | 0.29       | 0.55       | 0.14      |
> > | Random Init (h3)      | Hedonic-PAS        | 0.17       | 0.20       | 0.05      |
> > | Label-Shuffled (h3)   | Hedonic-PAS        | 0.19       | 0.23       | 0.06      |
> >
> > ---
> >
> > ## Conclusion
> >
> > The MNIST experiments demonstrate that:
> >
> > 1. Our method is **not** LLM-specific.
> > 2. Hedonic-PAS coalitions are **functionally meaningful**, with large accuracy drops and high synergy.
> > 3. Hedonic-PAS coalitions are **highly interpretable**, consistently capturing stroke-level visual motifs.
> > 4. Baselines fail to recover coherent structure, confirming that **synergistic coalition formation** is crucial.

---

### Author Response · Authors · 2025-12-03
**Author's Summary of Reviews and Rebuttal**

We thank all reviewers for their thoughtful feedback. Reviewers highlighted the **novelty** of our *hedonic game-theoretic framework* for modeling neurons as cooperative agents (dFvZ, tscc, SAy9), noting that it provides a new and computationally feasible bridge between cooperative game theory and neural network analysis (dFvZ). They emphasized that *Hedonic-PAS coalitions* are **functionally essential**, with ablations producing substantially larger performance drops than similarity- or correlation-based clustering baselines, and that the coalitions align with known IR heuristics and meaningful cross-layer dynamics (dFvZ, tscc). Reviewers also praised the **interpretability, theoretical soundness, and clarity** of the approach, especially for understanding challenging MLP-layer structure (tscc, SAy9), and appreciated the overall **reproducibility and strong empirical support** provided in the paper (dFvZ, SAy9).

This is a summary of the main concerns raised in the reviews and how we addressed them:

1. **"Can we infer human understandable insights/vizualizations to validate the value of coalitions" (dFvZ):**
We added a new experiment with the MNIST dataset (Appendix I) to visualize the effectiveness of the computed neuron coalitions. We introduced a **stroke-alignment metric** using simple edge detectors corresponding to canonical MNIST strokes (top bars, loops, vertical stems, diagonals). PAS coalitions achieve substantially higher alignment scores than all baselines, and we illustrate their visual coherence with **top-activating image grids** (Appendix I Figures 3–6).

2.  **"Does the method generalize beyond LLM backbones (dFvZ), LoRA tuning (dFvZ, tscc, SAy9), and IR tasks?":**
To directly address this concern, we added a new experiment on a **non-LLM 6-layer Gated MLP trained on MNIST** from scratch. We compute hedonic coalitions across all neurons in every internal layer, and show that PAS-based coalitions recover clear stroke-level visual concepts (loops, horizontal bars, diagonals). Hedonic-PAS significantly outperforms random partitions, k-means, and hierarchical clustering on **digit purity, digit selectivity, sparsity, synergy, and ablation accuracy drop**. These results are presented in Appendix I, with key quantitative findings summarized in Tables 7–9.

With the above points, combined with detailed rebuttal, we believe we have addressed the major concerns raised by reviewers.

Importantly, reviewers note that this work introduces a **novel game-theoretic framing** of mechanistic interpretability (dFvZ, tscc, SAy9), and that Hedonic-MFC with PAS valuations identifies **functionally essential** and **conceptually meaningful** neuron groups that outperform similarity-based baselines across tasks and architectures (dFvZ, tscc).

---

**NOTE** : We emphasize that the primary goal of our framework is to reduce the massive 14k–neuron space into a small number of meaningful, stable groups that future work can reverse-engineer. Interpreting every coalition is an open problem, and we explicitly list this as future work. We only give evidence hinting towards the usefulness of our discovered coalitions in the rebuttal. Such evidence is only possible to find in visual tasks like mnist. The end goal is to mathematically reverse engineer each coalition in future work, which is an important open problem in the community and beyond the scope of this work.

---

### Meta-Review · Area_Chair_s7hP · 2026-01-06

**Summary:**

The authors introduce a novel game-theoretic approach for identifying groups/coalitions of Neurons within Transformer MLP blocks based on the synergy of their interactions to layer-local computations. The functional importance of such coalitions is identified by measuring performance drops after resetting such Neurons to their pre-update state compared to various baseline selection methods. Coalitions themselves are identified through hedonic coalition formation

The authors use tools from game theory, specifically hedonic coalition formation and the PAC-Top-Cover algorithm, to identify these “neuron coalitions” and show that they are functionally important by measuring performance drops after removing these neurons.

The submission's clear strength lies in the novelty and theoretical soundness of its methodological framework, which all reviewers acknowledge. Remaining (post-rebuttal) concerns around the scale and diversity of tasks, as well as the generalisability of findings, are valid, but in my mind, they are setting an unrealistic expectation of the level of contributions that ought to be expected from an ICLR conference paper. It is indeed the case that the average ICLR paper may have more diverse and extensive experimental evaluation, but it is equally true that the average paper presents far less methodological novelty than this submission. As a community, I feel strongly that we ought to value both types of contributions (empirically strong vs methodologically strong) to the conference.

As the authors have done their best to provide additional results during the rebuttal, which I feel would have increased the score of Reviewer dFvZ, I would recommend acceptance. That being said, I would require the authors to clearly state that the method "is designed mainly for scalar-output tasks" in the Appendix, as this clearly indicates the early stage of this research direction. At the moment, this is not clear, and the choice of scalar IR tasks appears coincidental at first.

**Reviewer Concerns:**

I feel that outstanding concerns are most clearly formulated by Reviewer SAy9: "The results and analysis are still exploratory", "narrow task scope may limit generalizability of the proposed framework" and "the results may not generalize (in more of a scalability sense) to other layers, architectures, or non-LoRA fine-tuning setups".

While the authors have partially addressed this by adding a non-LoRA, non-LLM MNIST classification task (which is appreciated but also arguably the simplest ML task in existence), I think it's clear that future work would be well advised to test this method in a much wider range of more complex and diverse settings.

**Reviewer Scores:**

I believe that Reviewer dFvZ would have likely increased their score from a 4->6, placing this submission in the Top 25% of papers in my assigned batch. The authors very clearly addressed all concerns; the reviewer admits low confidence, and additional experimental results have been provided.

---

### Decision · Program_Chairs · 2026-01-26

Accept (Poster)